# WorldTree: Towards 4D Dynamic Worlds from Monocular Video using Tree-Chains

**Qisen Wang, Yifan Zhao,**[*] **Jia Li**
State Key Laboratory of Virtual Reality Technology and Systems
School of Computer Science and Engineering & Qingdao Research Institute, Beihang University
{wangqisen, zhaoyf, jiali}@buaa.edu.cn

## Abstract

Dynamic reconstruction has achieved remarkable progress, but there remain challenges in monocular input for more practical applications. The prevailing works attempt to construct efficient motion representations, but lack a unified spatiotemporal decomposition framework, suffering from either *holistic **temporal** optimization* or *coupled hierarchical **spatial** composition*. To this end, we propose **WorldTree**, a unified framework comprising Temporal Partition Tree (**TPT**) that enables coarse-to-fine optimization based on the inheritance-based partition tree structure for hierarchical temporal decomposition, and Spatial Ancestral Chains (**SAC**) that recursively query ancestral hierarchical structure to provide complementary spatial dynamics while specializing motion representations across ancestral nodes. Experimental results on different datasets indicate that our proposed method achieves 8.26% improvement of LPIPS on NVIDIA-LS and 9.09% improvement of mLPIPS on DyCheck compared to the second-best method. Code: https://github.com/iCVTEAM/WorldTree.

## 1 Introduction

Novel View Synthesis (NVS) has achieved remarkable progress in photo-realistic rendering, *e.g.* Neural Radiance Field and 3D Gaussian Splatting. Recently, the dynamic NVS, also known as dynamic 3D reconstruction, has achieved rapid advancements, extending the paradigm to temporally varying scenes by leveraging synchronized multi-view video inputs for novel view rendering across arbitrary timestamps. However, significant challenges remain in monocular dynamic reconstruction, which could reduce the dependency on synchronized multi-view capturing and enable more practical applications, given the abundance of available monocular videos.

Although the prevailing monocular reconstruction methods have demonstrated significant progress in developing efficient motion representations, there remains a notable gap in sufficiently analyzing the video modality in the reconstruction process. An intuitive analysis of previous methods is shown in Fig. 1. First, the prevailing methods (Lei et al., 2025; Wang et al., 2024b) commonly adopt *global optimization*, while disregarding temporal characteristics in video modality and relying on the optimization across the entire temporal interval. Furthermore, some researchers (Lei et al., 2025) attempt to utilize spatial motion graphs with global spatial fusion but introduce the *amorphous representation* and neglect temporal aspects. Others (Liang et al., 2025) decompose 3D deformations into global-local hierarchies, yet introduce *hierarchical motion coupling*. Inspired by the issues encountered with prevailing methods, we aim to develop a unified optimization pipeline that adapts to the *spatiotemporal characteristics of video modality*.

Intuitively, monocular videos exhibit varying deformation patterns across temporal intervals, stemming from non-uniform 3D motion of the subject. Furthermore, the deformation characteristics of any temporal interval inherit properties from its parent temporal interval, which simultaneously encompasses the spatial complement to the current interval. The above analysis reveals an inherent spatiotemporal hierarchy, *i.e.* a *tree-like structure with ancestral associations*. To this end, we propose **WorldTree**, which includes the *Temporal* Partition Tree (TPT) and the *Spatial* Ancestral

---

[*]Correspondence should be addressed to Yifan Zhao

Chains (SAC) as shown in Fig. 1. Existing methods employ entire-interval-wide optimization (Lei et al., 2025), whereas TPT implements hierarchical temporal partitioning to handle varying deformation patterns. For dynamic representation, existing methods rely on global spatial fusion with amorphous representation (Lei et al., 2025) or hierarchical motion composition (Liang et al., 2025), struggling with hierarchical deformation optimization coupling, whereas SAC recursively utilizes hierarchical spatial composition with ancestral specialization to circumvent amorphous representations and optimization coupling.

Specifically, **TPT** decomposes the monocular sequence into hierarchically organized temporal intervals through sequential layering, thereby constructing an inheritance-based partition tree. Additionally, we utilize the attribute independence among nodes at the same hierarchical level to enable parallel optimization within the same layer. For each child node in the tree, **SAC** recursively retrieves its ancestor chain to obtain hierarchical representations. The node itself models local temporal dynamics, while its ancestral chain provides multi-level spatial context. Distinctively, our method specializes the motion representation of each ancestral node to achieve hierarchical motion decoupling. Furthermore, we construct an enhanced **NVIDIA-LS** (Yoon et al., 2020) dataset

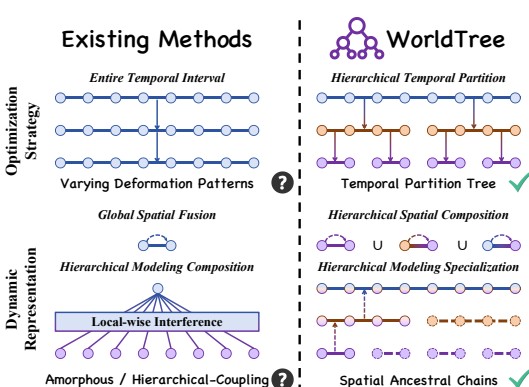

Figure 1: Intuitive illustration of WorldTree.

featuring *extended temporal length* with varying deformation patterns. To enable more precise quality assessment of dynamic reconstruction, we additionally provide *annotated motion masks*.

Our contributions can be summarized as

- *Temporal Partition Tree* builds the inheritance-based partition tree-structure by adopting the coarse-to-fine optimization strategy with hierarchical temporal partition to cope with the varying deformation patterns of monocular videos.

- *Spatial Ancestral Chains* provides a spatially complementary dynamic representation and introduces hierarchical node specialization on ancestral chains, thereby achieving optimization decoupling while constructing the hierarchical complementary representation.

- *Enhanced Dataset* with extended temporal length, indicating varying motion deformation patterns, and the annotated dynamic masks for precise evaluation.

- *High-quality Reconstruction*. Experimental results on different datasets show that our proposed method achieves $8.26\%$ improvement of LPIPS on NVIDIA-LS and $9.09\%$ improvement of mLPIPS on DyCheck compared to the second-best method.

## 2 RELATED WORK

**Novel View Synthesis** Novel View Synthesis(NVS) (Avidan & Shashua, 1997) focuses on reconstructing the 3D representations for rendering images on novel views given a set of captured training images. Neural Radiance Field (NeRF) (Mildenhall et al., 2021; Rabby & Zhang, 2023; Gao et al., 2022b; Barron et al., 2021; Müller et al., 2022; Chen et al., 2022), as a milestone in the development of NVS, implicitly represents the 3D scene using the Multi-Layer Perceptron (MLP), and achieves photo-realistic rendering quality. However, its speed of training and rendering is usually slow. 3D Gaussian Splatting (3DGS) (Kerbl et al., 2023; Chen & Wang, 2024; Fei et al., 2024; Yu et al., 2024), as a recently emerged 3D representation, explicitly utilizes a set of Gaussian primitives to model the 3D scene, and achieves real-time rendering and high-quality reconstruction through differentiable splatting rendering. Although previous 3D reconstruction work can handle static scenes well, it cannot efficiently handle dynamic scenes.

**Dynamic Novel View Synthesis** Dynamic NVS (Zitnick et al., 2004; Stich et al., 2008; Gao et al., 2021; Li et al., 2023b; 2022) has recently attracted more attention, which aims to reconstruct the

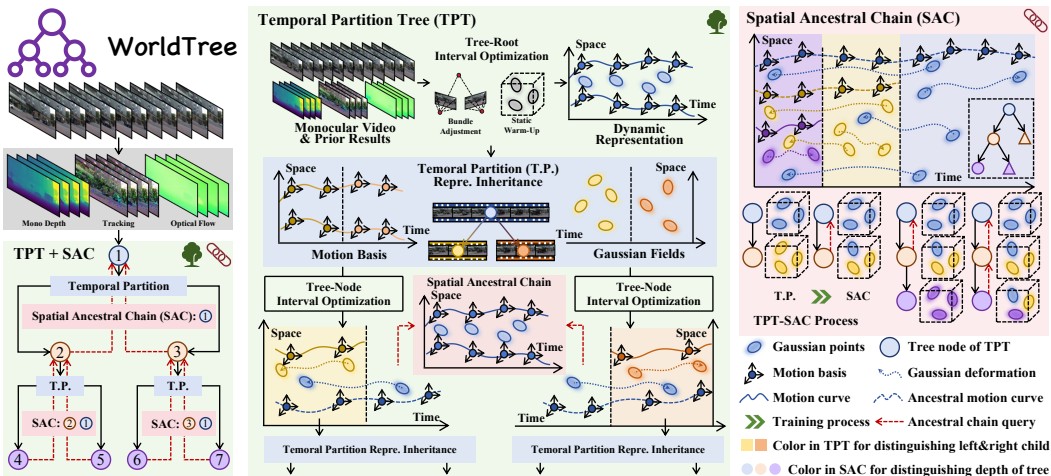

Figure 2: WorldTree Pipeline. Our proposed method starts by extracting 2D prior results and then initializes the dynamic representation of the tree root. Furthermore, WorldTree builds TPT to achieve temporal coarse-to-fine optimization from the overall interval to the sub-interval of the video, and utilizes SAC to achieve the complementary spatial dynamic representation at the same time, thereby achieving high-quality dynamic reconstruction.

4D representation of the dynamic 3D scene given a set of time-synchronized videos, and further render images of novel views at different timestamps (Attal et al., 2023; Shao et al., 2022; Liu et al., 2022; Guo et al., 2022; Song et al., 2023). Similar to the static NVS, the previous works are mainly modeled based on NeRF. The above methods implicitly construct the deformation field through the mapping between frames and the canonical space, using neural networks for modeling the captured motions. Recently, due to the rapid development of 3DGS, dynamic scene modeling (Duan et al., 2024; Lin et al., 2023; Li et al., 2023a; Huang et al., 2023; Shao et al., 2024) has been migrated to be 3DGS-based and achieved faster rendering speed (Duisterhof et al., 2023; Liang et al., 2023; Yang et al., 2023a). The 3DGS-based methods usually have faster training and rendering speeds, but like the NeRF-based methods (Cao & Johnson, 2023; Fridovich-Keil et al., 2023), they require time-synchronized multi-view data, which generally needs to be shot in a specific scene or requires a certain number of cameras, thus limiting the scope of application.

**Monocular Dynamic Novel View Synthesis** As mentioned above, time-synchronized multi-view videos are rare in the real world, but monocular causal videos are massive. Therefore, how to use monocular videos with few constraints for high-quality dynamic reconstruction becomes an important challenge (Liu et al., 2024; Seidenschwarz et al., 2024; Yang et al., 2021; You & Hou, 2023; Wu et al., 2022; Tretschk et al., 2021; Das et al., 2023). The majority of previous works are limited by the dependency on the multi-view clues, *e.g.* the utilization of pre-computed initialized points (Wu et al., 2023; Yang et al., 2023b) from SfM (Schonberger & Frahm, 2016). Recent work by (Lei et al., 2025) employs spatial graphs of motion bases for monocular dynamic reconstruction, yet produces an amorphous dynamic representation. Some work (Wu et al., 2025) attempts streamable 4D modeling from monocular video, but lacks an explicit mechanism for coarse-to-fine temporal partition optimization across the entire sequence. The other work (Liang et al., 2025) decomposes 3D motion deformations into global-local hierarchies, but their reliance on precise coarse deformation estimation makes them vulnerable to optimization conflicts by local-wise interference, ultimately causing hierarchical motion coupling. Additionally, some researchers (Kwak et al., 2025) implement temporal splitting by adaptively adjusting temporal intervals, but this division relies on the manual setting of temporal segments and has limited improvement on reconstruction quality. To this end, we propose WorldTree for the monocular dynamic reconstruction as demonstrated in Sec. 1.

## 3 METHOD

In this section, keeping the analysis of Sec. 1 for monocular dynamic reconstruction in our mind, we propose WorldTree for this task. Given one monocular video including a set of images $\mathbb{I} =$

$\{I_t\}_{t \in [1,T]}$, and its corresponding poses $\mathbb{P} = \{P_t\}_{t \in [1,T]}$, where $T$ is the video length, we aim to reconstruct the dynamic scene, which includes the static background and moving foreground objects, namely the monocular dynamic reconstruction. The overview of our pipeline is shown in Fig. 2, which includes two main parts, *i.e.*, Temporal Partition Tree (TPT) and Spatial Ancestral Chains (SAC). We first introduce the basic formulation of monocular dynamic reconstruction from the previous work (Lei et al., 2025) while defining the core symbols, and then we demonstrate TPT and SAC.

### 3.1 MONOCULAR DYNAMIC RECONSTRUCTION

**Lifting 2D Priors.** Similar to previous works (Wang et al., 2024b; Lei et al., 2025), we first lift several 2D pretrained priors to the 3D scene, which needs to be reconstructed. Specifically, given the monocular images $\mathbb{I}$, we compute depths $\mathbb{D} = \{D_t\}_{t \in [1,T]}$ from the monocular depth predictor, 2D points trajectories $\mathbb{T} = \{p_t^i\}_{t \in [1,T]}$ from the point tracking model where $p_t^i$ means pixel coordinates and $i$ represents $i$-th trajectory, and epipolar error maps $\mathbb{E} = \{E_t\}_{t \in [1,T]}$ from the dense optical flow model. The epipolar error maps can be further utilized to compute the EPI masks, which do not need manual point prompts (Wang et al., 2024b; Park et al., 2025; Liang et al., 2025). These predicted results from 2D priors are utilized to initialize the representation of the dynamic scene for further optimization.

**Deformation Representation.** The deformation is represented as a set of motion basis $\mathcal{M} = \{\mathbf{M}_t^{(\psi)}\}_{t \in [1,T]}$ with radius $r^{(\psi)}$, where $\psi \in [1, N_m]$, and $\mathbf{M}_t^{(\psi)} = [\mathbf{R}_t^{(\psi)}, \mathbf{T}_t^{(\psi)}] \in SE(3)$. For each motion basis $\psi$, we compute its KNN graph $\mathcal{G}(\psi) = \{\mathbf{M}^{(\eta)}, \mathbf{E}^{(\psi)}\}$, whose

$$\mathbf{E}^{(\psi)} = \mathrm{KNN}_{\eta \in [1, N_m]}\{\mathcal{D}(\psi, \eta)\}, \tag{1}$$

where the distance $\mathcal{D}(\psi, \eta)$ is the maximum spatial distance across timestamps. The deformation of motions can be represented as the relative deformation of the motion bases. Specifically, given a query of position $\mathbf{T}_{t_s}^{(x)}$ from time $t_s$ to $t_d$, the nearest motion basis $\psi_x$ at time $t_s$ is $\mathbf{M}_{t_s}^{(\psi_x)}$ and the deformation field $\mathcal{W}(\mathbf{T}_{t_s}^{(x)}, \mathcal{G}(\psi_x), t_s, t_d)$ is defined as

$$\mathcal{F}_B(\{\Delta\mathbf{M}_{t_s \to t_d}^{\eta_x}, w_{\eta_x}(\mathbf{T}_{t_s}^{(x)}, \mathbf{T}_{t_s}^{(\eta_x)}, r^{(\eta_x)})\}_{\eta_x \in \mathbf{E}^{(\psi_x)}}), \tag{2}$$

where $\Delta\mathbf{M}_{t_s \to t_d}^{\eta_x} = \mathbf{M}_{t_d}^{\eta_x}(\mathbf{M}_{t_s}^{\eta_x})^{-1}$, $\mathcal{F}_B$ is blending function which is set to Dual Quaternion Blending (DQB), and $w_{\eta_x}(\mathbf{T}_{t_s}^{(x)}, \mathbf{T}_{t_s}^{(\eta_x)}, r^{(\eta_x)}) = \exp(-\|\mathbf{T}_{t_s}^{(x)} - \mathbf{T}_{t_s}^{(\eta_x)}\|_2^2 / 2r^{(\eta_x)})$.

**Dynamic Reconstruction.** Given the modeling of the deformation field, the dynamic reconstruction can be represented as a set of dynamic Gaussians in the canonical space $\mathbb{G} = \{G_n(x_n, R_n, s_n, o_n, c_n, t_n)\}$, which represents the position, rotation, scaling, opacity, color, and canonical time of Gaussian $n$ respectively, with the above mentioned deformation fields. Specifically, the deformation of the Gaussian $n$ to target time $\tau$ in the video interval can be represented as

$$G_n(\mathcal{M}, \tau) = (\mathcal{W}(\mathbf{T}_{t_n}^{(\psi_{x_n})}, \mathcal{G}(\psi_{x_n}), t_n, \tau)x_n,$$
$$\mathcal{W}(\mathbf{T}_{t_n}^{(\psi_{x_n})}, \mathcal{G}(\psi_{x_n}), t_n, \tau)R_n, s_n, o_n, c_n, \tau). \tag{3}$$

So, given the Gaussians $\mathbb{G}$ in the canonical space and the deformation field $\mathcal{W}(\cdot)$ constructed by $\mathcal{G}$ from $\mathcal{M}$, we can model the dynamic scene through the splatting way. However, although the above modeling provides efficient motion representation, it lacks optimization based on temporal characteristics and is limited in spatial representations by the global optimization.

### 3.2 TEMPORAL PARTITION TREE

As we demonstrated in Sec. 1, the classical paradigm of monocular dynamic reconstruction mentioned above (Lei et al., 2025) processes the video sequence as a whole and lacks hierarchical modeling optimization of time and space for dynamic scenes. To this end, we propose WorldTree for this issue. We first introduce the Temporal Partition Tree (TPT).

Specifically, we start from the tree root $\zeta_r(\mathcal{M}_r, \mathbb{G}_r)$ of tree-depth $d_r = 0$ and representing the whole video interval $[T_r^L, T_r^R]$ obtained from the above reconstruction, which represents the basic

---

**Algorithm 1** BFS-Travel Tree-Building

---

**Input**: Tree root $\zeta_r$ with motion bases $\mathcal{M}_r$, Gaussian representation $\mathbb{G}_r$. Empty SAC $\mathcal{C}_r = []$. Max depth $\delta$. Initial depth $d_r = 0$.
**Output**: WorldTree $\mathcal{Q}$.

1: Create a tree $\mathcal{Q}$, and $\mathcal{Q}.\text{set}(r = 1; \zeta_r(\mathcal{M}_r, \mathbb{G}_r); \mathcal{C}_r)$.
2: **for** $d_j \in [d_r, \delta]$ **do**
3:     Parallel training for $\{(j; \zeta_j(\mathcal{M}_j, \mathbb{G}_j); \mathcal{C}_j)\} \leftarrow \mathcal{Q}.\text{get}(2^{d_j}, 2^{d_j+1} - 1)$.
4:     **for** All $j$ **do**
5:        $\mathcal{M}_j, \mathbb{G}_j$ temporally partitioned to $\mathcal{M}_{2j}, \mathbb{G}_{2j}$ (Left), and $\mathcal{M}_{2j+1}, \mathbb{G}_{2j+1}$ (Right).
6:        Left: $\mathcal{C}_{2j} = \text{Append}(\mathcal{C}_j, \zeta_j)$. Right: $\mathcal{C}_{2j+1} = \text{Append}(\mathcal{C}_j, \zeta_j)$.
7:        Left: $\mathcal{Q}.\text{set}(2j; \zeta_{2j}(\mathcal{M}_{2j}, \mathbb{G}_{2j}); \mathcal{C}_{2j})$. Right: $\mathcal{Q}.\text{set}(2j + 1; \zeta_{2j+1}(\mathcal{M}_{2j+1}, \mathbb{G}_{2j+1}); \mathcal{C}_{2j+1})$.
8:     **end for**
9: **end for**

---

and coarse dynamic modeling and includes the motion bases $\mathcal{M}_r$ and the dynamic Gaussian representation $\mathbb{G}_r$. Note that $r$ is set to 1 for the root. We further build this partition tree using BFS traveling, which means that as the depth of the tree expands, the temporal intervals become narrower and more refined. For each newly added tree node $\zeta_j(\mathcal{M}_j, \mathbb{G}_j)$, we use a similar training process as the root node for optimization to achieve the refinement of the temporal interval. Then, the optimized tree node for the video interval $[T_j^L, T_j^R]$ is partitioned into left and right child nodes according to the temporal partition point, namely $[T_j^L, T_j^P]$ for the left child $\zeta_{2j}(\mathcal{M}_{2j}, \mathbb{G}_{2j})$, and $[T_j^P, T_j^R]$ for the right child $\zeta_{2j+1}(\mathcal{M}_{2j+1}, \mathbb{G}_{2j+1})$, where $T_j^P = \lfloor (T_j^L + T_j^R)/2 \rfloor$ is the partition point which we obtain using the binary strategy. The partition process includes the division of the motion bases

$$\mathcal{M}_k = \begin{cases} \mathcal{M}_{2j} = \left\{ \mathbf{M}_t^{(\psi)} \in \mathcal{M}_j \mid t < T_j^P \right\}, \\ \mathcal{M}_{2j+1} = \left\{ \mathbf{M}_t^{(\psi)} \in \mathcal{M}_j \mid t \geq T_j^P \right\}, \end{cases} \quad (4)$$

and the division of the Gaussian primitives

$$\mathbb{G}_k = \begin{cases} \mathbb{G}_{2j} = \left\{ G_n^j \in \mathbb{G}_j \mid t_n^j < T_j^P \right\}, \\ \mathbb{G}_{2j+1} = \left\{ G_n^j \in \mathbb{G}_j \mid t_n^j \geq T_j^P \right\}. \end{cases} \quad (5)$$

In practice, we limit the number of Gaussian primitives that a child node inherits from its ancestor to balance efficiency and reset its opacity before training to escape from local saddle points.

### 3.3 SPATIAL ANCESTRAL CHAINS

Although the video interval has been partitioned to achieve the temporary coarse-to-fine dynamic modeling optimization, the partition of the temporal interval lacks the spatial visual supplement, because the Gaussian primitives are constantly truncated during the inheritance process. Therefore, we further propose the Spatial Ancestral Chains (SAC) to introduce dynamic representations at different spatial scales.

Specifically, for each tree node $\zeta_j(\mathcal{M}_j, \mathbb{G}_j)$, we introduce the dynamic expression chain $\mathcal{C}_j$ from the root node ancestor to this node, which can be represented as

$$\mathcal{C}_j = \left\{ \zeta_k^j(\mathcal{M}_k^j, \mathbb{G}_k^j) \mid k \in \left\lfloor \frac{j}{2^\alpha} \right\rfloor, \alpha = 1 \ldots \log_2 j \right\}, \quad (6)$$

where $\alpha$ represents the number of upstream levels related to $j$, and $\zeta_k^j(\mathcal{M}_k^j, \mathbb{G}_k^j)$ means that the common ancestors of different nodes are independent of each other but have the same optimized initialization. Note that the dynamic representation in $\mathcal{C}_j$ is also optimized in the process of building TPT. Thus, we construct a spatial hierarchical representation of the video interval $[T_j^L, T_j^R]$ with hierarchical spatial composition and modeling specialization. Furthermore, the final deformed Gaussian points $\mathcal{P}_j$ at time $\tau$ can be written as

$$\mathcal{P}_j = \left\{ G_n^j(\mathcal{M}_j, \tau) \mid G_n^j \in \mathbb{G}_j \right\}_{\zeta_j(\mathcal{M}_j, \mathbb{G}_j)} \cup$$
$$\left\{ \cup_k \left\{ G_n^k(\mathcal{M}_k^j, \tau) \mid G_n^k \in \mathbb{G}_k^j \right\}_{\zeta_k^j(\mathcal{M}_k^j, \mathbb{G}_k^j) \in \mathcal{C}_j} \right\}, \quad (7)$$

The following rasterization process is similar to the common splatting process. By introducing SAC, we introduce motion representations from different spatial aspects to further assist the dynamic modeling of the current node representing the partial video interval.

## 3.4 TRAINING DETAILS

**Parallel Optimization.** An intuitive observation is that although nodes at the same depth have the same ancestor chain initialization, they are independent of each other during the optimization process. Therefore, we provide parallel optimization of the same depth and achieve $O(\delta)$ optimization efficiency in terms of optimization times when there are enough parallel computing resources. If the optimization is not performed in parallel, $O(2^{\delta+1})$ times of independent optimization are required.

**Root-Optimization.** We adopt several pre-optimization processes for the root node to better construct the dynamic representation during the overall training process. **Bundle Adjustment.** Since it is often difficult to obtain sufficiently accurate camera poses in dynamic reconstruction, we additionally adopt tracklet-based bundle adjustment following (Lei et al., 2025) to better initialize the camera before dynamic optimization. **Static Regions Warm-up.** In order to make the model focus more on dynamic modeling optimization during dynamic foreground reconstruction, we pre-optimize the static background before dynamic reconstruction to obtain a stable static background as a warm-up for the static regions of the overall dynamic reconstruction.

**Photometric Optimization with WorldTree.** In Algo. 1, we demonstrate the algorithmic core process of the proposed method. This algorithmic process primarily describes the optimization strategy and its structure, which are the core content of this work. More specifically, when training each node, we use several regularization terms to achieve comprehensive optimization, including regularizations commonly used in the prevailing methods (Lei et al., 2025; Park et al., 2025) such as photometric loss and depth loss. See more implementation details in the Appendix.

## 3.5 DISSCUSION

We provide a detailed discussion in comparison to other prevailing methods. *Temporal Partition*. MoDec-GS (Kwak et al., 2025) attempts to achieve efficient motion representation through adaptive time intervals and global-local decomposition of motion, but is limited to fixed partition initialization and two-stage optimization. *Hierarchical Representation*. MoSca (Lei et al., 2025) introduces an efficient motion scaffold graph for fusing the motions, but its optimization of global deformation in the overall interval and non-hierarchical motion representation limit its modeling capabilities. HiMoR (Liang et al., 2025) proposes the hierarchical motion representation with a tree structure for decomposing motions to coarse-to-fine levels, but neglects hierarchical-coupling of optimization conflicts across different temporal intervals with varying deformation patterns caused by the local-wise interferences. *Manual Point Prompts*. Besides, the prevailing works (Park et al., 2025; Wang et al., 2024b; Liang et al., 2025) tend to introduce the manual point prompts prior to accurately identifying the dynamic regions for better reconstruction, but the manual points conflict with the massive amount of monocular videos, which is further discussed in Sec. 4. To this end, we propose the WorldTree for both achieving hierarchical temporal partitioning for monocular video intervals and spatial complementary hierarchical dynamic representations with modeling specialization.

## 4 EXPERIMENTS

### 4.1 EXPERIMENTAL DETAILS

**Dataset.** We utilize two datasets for evaluating the proposed method. We first re-construct the NVIDIA-LS (Yoon et al., 2020) dataset, which is more challenging than the original NVIDIA (Yoon et al., 2020) dataset. Specifically, Yoon et al. uses a rig of 12 cameras for capturing different views at the same timestamp, so that they construct a monocular video while obtaining poses from different views. The original dataset only contains 12 frames. We extend the video interval to a long sequence of up to 160 frames. Besides, the previous data division uses the fixed pose corresponding to the first frame of the training view as the test view, which implicitly reduces the difficulty of dynamic reconstruction of monocular video during training. Therefore, we separate the test view from the

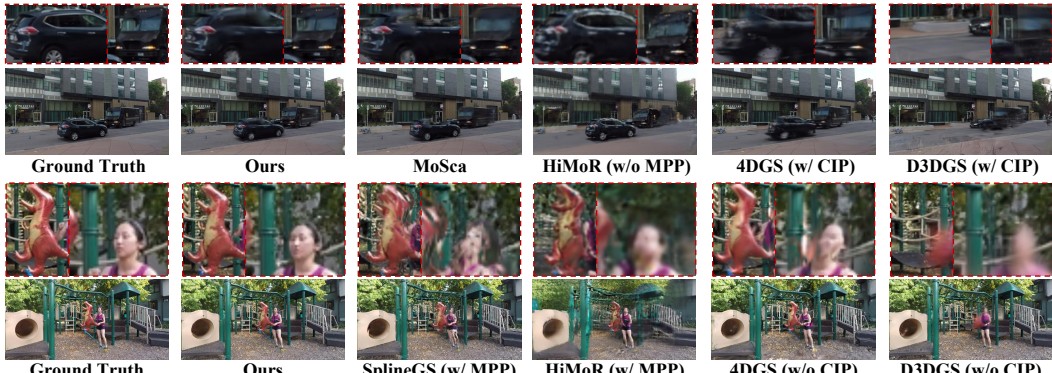

Figure 3: Qualitative comparisons with other methods on NVIDIA-LS (Yoon et al., 2020).

Table 1: **Quantitative comparisons** with other methods on NVIDIA-LS (Yoon et al., 2020).

| Method | CIP | MPP | PSNR↑ | SSIM↑ | LPIPS↓ | AVGE↓ | mPSNR↑ | mSSIM↑ | mAVGE↓ |
|---|---|---|---|---|---|---|---|---|---|
| D3DGS* † (Yang et al., 2023b) | ✗ | ✗ | 21.41 | 0.748 | 0.209 | 0.091 | 16.29 | 0.575 | 0.147 |
| D3DGS † (Yang et al., 2023b) | ✓ | ✗ | 21.33 | 0.728 | 0.228 | 0.096 | 16.19 | 0.563 | 0.154 |
| HiMoR* † (Liang et al., 2025) | ✗ | ✗ | 19.24 | 0.444 | 0.465 | 0.160 | 15.17 | 0.509 | 0.215 |
| HiMoR † (Liang et al., 2025) | ✗ | ✓ | 19.30 | 0.442 | 0.440 | 0.157 | 15.28 | 0.510 | 0.209 |
| SplineGS* † (Park et al., 2025) | ✗ | ✗ | 21.41 | 0.687 | 0.227 | 0.097 | 15.56 | 0.541 | 0.162 |
| SplineGS † (Park et al., 2025) | ✗ | ✓ | 22.20 | 0.725 | 0.168 | 0.081 | 17.07 | 0.617 | 0.127 |
| 4DGS* † (Wu et al., 2023) | ✗ | ✗ | 21.48 | 0.646 | 0.197 | 0.094 | 16.87 | 0.604 | 0.137 |
| 4DGS † (Wu et al., 2023) | ✓ | ✗ | 23.34 | 0.790 | 0.128 | 0.065 | 17.95 | 0.651 | 0.107 |
| MoSca † (Lei et al., 2025) | ✗ | ✗ | 23.72 | 0.792 | 0.109 | 0.060 | 17.89 | 0.664 | 0.101 |
| WorldTree (Ours) | ✗ | ✗ | **24.06** | **0.804** | **0.100** | **0.056** | **18.55** | **0.692** | **0.092** |

†: Our reproduction. *: Variants of methods. CIP: Colmap Initial Points. MPP: Manual Points Prompts.

training view, use the first camera as the test view, and construct the training data structure of the remaining cameras in a similar way to the original dataset. Furthermore, we annotate dynamic masks using Segment-Anything 2 (SAM2) (Ravi et al., 2024) for testing images to better evaluate the reconstruction quality of dynamic regions. See more details of the NVIDIA-LS (Yoon et al., 2020) in the Appendix. Note that SAM2 (Ravi et al., 2024) is only utilized to annotate testing images for evaluation, and not utilized in the optimization process. We also evaluate our method on the widely used DyCheck (Gao et al., 2022a) dataset following the protocol of the previous work (Lei et al., 2025).

**Implementation Details.** The preprocessing of our method using 2D fundamental models follows MoSca (Lei et al., 2025). We use Metric3D-V2 (Hu et al., 2024a) for monocular depth prediction, CoTracker3 (Karaev et al., 2023; 2024) for tracking, and RAFT (Teed & Deng, 2020) for obtaining optical flow. Note that the LiDAR-sensor depth is used for the DyCheck (Gao et al., 2022a) dataset following the previous works (Lei et al., 2025). The tree-depth of TPT is set to 2, which counts from 0. See more implementation details in the Appendix.

**Baselines.** For the NVIDIA-LS (Yoon et al., 2020) dataset, we compare our method with several state-of-the-art methods, including D3DGS (Yang et al., 2023b), 4DGS (Wu et al., 2023), SplineGS (Park et al., 2025), MoSca (Lei et al., 2025). It is supposed to be noted that some of the above methods (Yang et al., 2023b; Wu et al., 2023) introduce SfM (Schonberger & Frahm, 2016) initialization point cloud from Colmap (Schonberger & Frahm, 2016) results, while some methods (Park et al., 2025; Liang et al., 2025) use manual point prompts as conditions to obtain foreground mask, where we use SAM2 (Ravi et al., 2024) for reproduction. For a fair comparison, we also report the experimental results without these priors. For the DyCheck (Gao et al., 2022a) dataset, the reported results are provided by the previous works (Lei et al., 2025; Liu et al., 2023). See details of reproduction in the Appendix.

**Metrics.** For NVIDIA-LS (Yoon et al., 2020), we report PSNR, SSIM, and LPIPS (Zhang et al., 2018) for evaluating the entire rendering, which includes both static and dynamic regions. We also

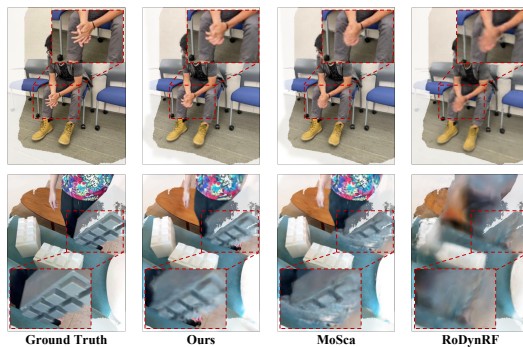

**Figure 4:** Qualitative comparisons with other methods on DyCheck (Gao et al., 2022a).

Table 2: **Quantitative comparisons** with other methods on DyCheck (Gao et al., 2022a).

| Method | mPSNR↑ | mSSIM↑ | mLPIPS↓ |
|---|---|---|---|
| T-NeRF (Gao et al., 2022a) | 16.96 | 0.577 | 0.379 |
| NSFF (Li et al., 2021) | 15.46 | 0.551 | 0.396 |
| Nerfies (Park et al., 2021a) | 16.45 | 0.570 | 0.339 |
| Hyp.NeRF (Park et al., 2021b) | 16.81 | 0.569 | 0.332 |
| PGDVS (Zhao et al., 2024) | 15.88 | 0.548 | 0.340 |
| DyPoint (Zhou et al., 2024) | 16.89 | 0.573 | - |
| DpDy (Wang et al., 2024a) | - | 0.559 | 0.516 |
| Dyn.Gau. (Luiten et al., 2023) | 7.29 | - | 0.692 |
| 4D GS (Wu et al., 2023) | 13.64 | - | 0.428 |
| DGMar. (Stearns et al., 2024) | 16.72 | - | 0.413 |
| DyBluRF (Bui et al., 2023) | 17.37 | 0.591 | 0.373 |
| CTNeRF (Miao et al., 2024) | 17.69 | 0.531 | - |
| D-NPC (Kappel et al., 2024) | 16.41 | 0.582 | 0.319 |
| RoDynRF (Liu et al., 2023) | 17.09 | 0.534 | - |
| SoM (Wang et al., 2024b) | 17.32 | 0.598 | 0.296 |
| MoSca (Lei et al., 2025) | 19.32 | 0.706 | 0.264 |
| WorldTree (Ours) | **19.75** | **0.728** | **0.240** |

report the masked PSNR (mPSNR) and masked SSIM (mSSIM) for evaluating the dynamic parts, which are more important for the dynamic reconstruction. Furthermore, we report the Average Error (AVGE), which is computed by the geometric mean of MSE $= 10^{-\text{PSNR}/10}$, $\sqrt{1 - \text{SSIM}}$, LPIPS, and masked Average Error (mAVGE), which replaces PSNR and SSIM with masked metrics. For the DyCheck (Gao et al., 2022a) dataset, we report mPSNR, mSSIM, and mLPIPS using the dataset-provided co-visibility masks following the previous works (Wang et al., 2024b; Lei et al., 2025).

## 4.2 COMPARISONS WITH OTHER METHODS

**NVIDIA-LS.** We conduct detailed *quantitative* experiments on the NVIDIA-LS (Yoon et al., 2020) dataset as shown in Tab. 1. Our method achieves the improvement of $21.40\%$ on the mP-SNR compared to HiMoR (Liang et al., 2025), the improvement of $12.16\%$ on the mSSIM compared to SplineGS (Park et al., 2025), and the improvement of $8.91\%$ on the mAVGE compared to MoSca (Lei et al., 2025). When replacing the mask manual prior using SAM2 (Ravi et al., 2024) with manual point prompts to the EPI mask, the improvement of our method goes up to $22.28\%$ on the mPSNR and $27.91\%$ on the mSSIM compared to HiMoR* (Liang et al., 2025) and SplineGS (Park et al., 2025), respectively. The *qualitative* results on the NVIDIA-LS (Yoon et al., 2020) dataset are shown in Fig. 3. Compared to the rendering results of SplineGS (Park et al., 2025) and HiMoR (Liang et al., 2025), the rendering results of our method show better overall rendering and reconstruction quality on the dynamic regions. Overall, our proposed method achieves both better reconstruction quality and more detailed visual content.

**DyCheck.** The *quantitative* experimental results on the DyCheck (Gao et al., 2022a) dataset are reported in Tab. 2. Similar to the results on the NVIDIA-LS (Yoon et al., 2020) dataset, our method achieves state-of-the-art results compared to prevailing methods. Following the evaluation protocol of the previous works (Liu et al., 2023; Lei et al., 2025), we evaluate the reconstruction quality using the masked metrics using the co-visibility masks. The mLPIPS of our method is improved by $18.92\%$ and $9.09\%$ compared to SoM (Wang et al., 2024b) and MoSca (Lei et al., 2025), respectively. The *qualitative* results on the DyCheck (Gao et al., 2022a) dataset are shown in Fig. 4. We can notice that MoSca (Lei et al., 2025), as the state-of-the-art method of 3DGS-based methods for dynamic reconstruction, still exhibits blurry rendering of details in motions of foreground objects, *e.g.* fingers in the top line of Fig. 4. See more qualitative comparisons in the Appendix.

## 4.3 PERFORMANCE ANALYSIS

**Why WorldTree achieves better reconstruction quality than prevailing methods?** As shown in 4.2, although the previous works (Lei et al., 2025; Wang et al., 2024b) propose efficient motion representations, their training strategy in the overall video sequences, and the global deformation field construction lead to their inadequacy in reconstructing motion details. Meanwhile, the current hierarchical representation (Liang et al., 2025) also struggles with hierarchical deformation coupling, thus achieving unsatisfactory reconstruction results. In contrast, our method achieves high-quality reconstruction of dynamic scenes. due to the *temporal coarse-to-fine optimization* strategy introduced by the proposed **TPT** and the *spatial hierarchical representation* capability with *chain-wise*

Table 3: **Ablation study** on the NVIDIA-LS (Yoon et al., 2020) dataset.

| BA | SW | TPT | SAC | Metric | | | | | | |
|----|----|-----|-----|--------|--|--|--|--|--|--|
| | | | | PSNR↑ | SSIM↑ | LPIPS↓ | AVGE↓ | mPSNR↑ | mSSIM↑ | mAVGE↓ |
| ✗ | ✗ | ✗ | ✗ | 21.87 | 0.710 | 0.139 | 0.079 | 16.82 | 0.620 | 0.121 |
| ✗ | ✗ | ✓ | ✗ | 22.39 | 0.777 | 0.121 | 0.069 | 17.41 | 0.652 | 0.109 |
| ✗ | ✗ | ✓ | ✓ | 22.64 | 0.778 | 0.113 | 0.066 | 17.85 | 0.668 | 0.102 |
| ✓ | ✗ | ✗ | ✗ | 22.08 | 0.712 | 0.130 | 0.076 | 17.32 | 0.642 | 0.113 |
| ✓ | ✗ | ✓ | ✗ | 22.58 | 0.764 | 0.115 | 0.068 | 18.02 | 0.673 | 0.101 |
| ✓ | ✗ | ✓ | ✓ | 22.73 | 0.766 | 0.108 | 0.065 | 18.35 | 0.685 | 0.096 |
| ✓ | ✓ | ✗ | ✗ | 23.59 | 0.787 | 0.115 | 0.061 | 17.73 | 0.655 | 0.104 |
| ✓ | ✓ | ✓ | ✗ | 23.94 | 0.801 | 0.105 | 0.058 | 18.36 | 0.683 | 0.095 |
| ✓ | ✓ | ✓ | ✓ | 24.06 | 0.804 | 0.100 | 0.056 | 18.55 | 0.692 | 0.092 |

(a) Quantitative comparisons of ablations on the NVIDIA-LS (Yoon et al., 2020) dataset.

(b) Qualitative comparisons of ablations on the NVIDIA-LS (Yoon et al., 2020) dataset.

Figure 5: Quantitative and qualitative ablations on the NVIDIA-LS (Yoon et al., 2020) dataset.

*specialization* introduced by the proposed **SAC**. Overall, our proposed method achieves both better reconstruction quality and more detailed visual content.

**Is WorldTree effective?** We conduct ablations on the NVIDIA-LS (Yoon et al., 2020) dataset shown in Tab. 3. Note that BA represents Bundle Adjustment, SW represents Static Warm-up, TPT represents Temporal Partition Tree, and SAC represents Spatial Ancestral Chains, which are demonstrated in Sec. 3. Our method achieves the improvement of $18.71\%$ on the LPIPS (the same below) when deactivating BA and SW, $16.92\%$ when deactivating SW, and $13.04\%$ when both activating BA and SW. This demonstrates the ability of WorldTree to optimize and promote the dynamic reconstruction in different situations, confirming its effectiveness.

**How does SAC promote TPT?** We also conduct ablations to explore the two hierarchical components of WorldTree: TPT and SAC, as shown in Tab. 3. When adding TPT, the LPIPS and mAVGE are improved by $8.70\%$ and $8.65\%$, respectively. When further adding SAC, the LPIPS and mAVGE are further improved by $4.76\%$ and $3.16\%$, respectively. This confirms that SAC further supplements the spatial hierarchical representation capability with chain-wise specialization based on TPT's temporal coarse-to-fine optimization. The detailed qualitative results of ablations are shown in Fig. 5b. The rendering quality corresponding to the progressive elimination of different parts of our final method demonstrates the effectiveness of each component. See more ablation results in the Appendix.

**How does the height of TPT affect the reconstruction quality?** The trend of reconstruction quality varying with TPT height is shown in Fig. 5a. We can notice that both TPT and SAC have an effective effect on improving reconstruction quality under different initializations. At the same time, the reconstruction quality is highly positively correlated with the height of the tree (*i.e.* the granularity of the time interval division) over several metrics. To balance quality and efficiency, we ultimately choose the maximum height $\delta$ of 2. More results and discussions on the tree height of TPT and the chain length of SAC are detailed in the Appendix.

**Exploration of different external priors.** To explore the sensitivity of our method to external priors, the experimental results under different external priors are shown in Tab. 4. While reconstruction quality varies with the initialization using depth and tracking priors, our method consistently achieves better reconstruction quality compared to MoSca (Lei et al., 2025), and achieves significant improvements, *e.g.*, $+10.27\%$ of mAVGE compared to Base when using Metric3D-V2 (Hu et al., 2024a) and BootsTAPIR (Doersch et al., 2024), and $+12.00\%$ of mAVGE compared to Base when using UniDepth (Piccinelli et al., 2024) and CoTracker3 (Karaev et al., 2023; 2024). The experimental results confirm the robustness of our method on different initializations of external priors.

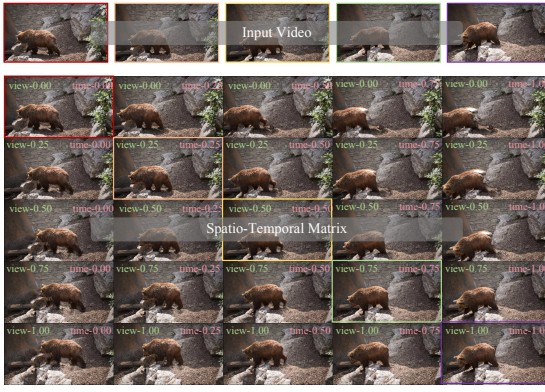

Figure 6: Qualitative results on the DAVIS (Pont-Tuset et al., 2017) dataset.

Table 4: Quantitative results of different external priors on NVIDIA-LS Yoon et al. (2020).

| Methods | Metric | | |
|---------|--------|--------|--------|
| | mPSNR↑ | mSSIM↑ | mAVGE↓ |
| *UniDepth + CoTracker3* | | | |
| MoSca | 18.30 | 0.674 | 0.097 |
| Base | 18.17 | 0.673 | 0.100 |
| Ours | **18.98** | **0.714** | **0.088** |
| *Metric3D-V2 + BootsTAPIR* | | | |
| MoSca | 16.94 | 0.609 | 0.142 |
| Base | 16.80 | 0.600 | 0.146 |
| Ours | **17.43** | **0.640** | **0.131** |
| *Metric3D-V2 + CoTracker3* | | | |
| MoSca | 17.89 | 0.664 | 0.101 |
| Base | 17.73 | 0.655 | 0.104 |
| Ours | **18.55** | **0.692** | **0.092** |

Table 5: Evaluation of VRAM costs.

| VRAM(GB) | Peak VRAM | Avg. VRAM |
|----------|-----------|-----------|
| MoSca | 23.99 | 12.90 |
| Ours (Layer 0) | 23.99 | 8.61 |
| Ours (Layer 1) | 23.99 | 11.99 |
| Ours (Layer 2) | 24.00 | 12.68 |

**Analysis of computational costs.** Furthermore, we compare the VRAM usage of our method and MoSca (Lei et al., 2025) during photometric optimization on the Balloon1 scene of the NVIDIA-LS (Yoon et al., 2020) dataset. As shown in Table 5, our method exhibits consistently lower average memory consumption across different tree layers compared to MoSca (Lei et al., 2025). The peak memory usage is similar for all methods, indicating a shared constraint inherited from the underlying architecture rather than being introduced by our Tree-Chains representation. Notably, our method maintains a significantly lower average VRAM footprint, demonstrating that Tree-Chains do not constitute a scalability bottleneck. Additional analysis of computational cost for parallel training is provided in the Appendix.

**Evaluation on the wild data.** To assess generalization on real-world videos, we present qualitative results on DAVIS (Pont-Tuset et al., 2017) as shown in Fig. 6. We select viewpoints at timestamps with the ratio of $[0, 0.25, 0.5, 0.75, 1.0]$ and render novel views to construct a spatio-temporal matrix. This matrix visualizes scene dynamics and 3D spatial information from multiple viewpoints at synchronized times. The consistent, high-quality renderings across the matrix confirm that our method generalizes effectively to in-the-wild videos, producing high-fidelity reconstructions and validating its practical applicability. More qualitative results on the wild data are provided in the Appendix.

## 5    CONCLUSION AND LIMITATIONS

In this paper, we propose WorldTree, which includes the Temporal Partition Tree and the Spatial Ancestral Chains. These two components implement the temporal progressive training strategy for the entire video interval and a complementary hierarchical representation at the spatial level. Furthermore, to better evaluate the quality of dynamic reconstruction, we propose the NVIDIA-LS dataset with a longer sequence length and annotated dynamic foreground masks for evaluation. Experimental results on different datasets show that our method achieves the state-of-the-art reconstruction quality. The limitation of our work is that the proposed pipeline still relies on the capability of pretrained foundation models. We can further improve the techniques for prior extraction. We leave it as our future work.

### ACKNOWLEDGMENTS

This work is partially supported by grants from the Guizhou Provincial Major Scientific and Technological Program (Qiankehe Zhongda [2025] No. 032), National Natural Science Foundation of China (No.62132002), Beijing Nova Program (No.20250484786), and the Fundamental Research Funds for the Central Universities.

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

## APPENDIX

## A USAGE OF LLMS

In this work, we use Large Language Models (LLMs) to aid or polish writing.

## B DATASET DETAILS

Prior to introducing our proposed NVIDIA-LS (Yoon et al., 2020) dataset, it is necessary to first discuss the limitations of the original NVIDIA (Yoon et al., 2020) dataset that motivated our reconstruction efforts. As illustrated in Fig. 7 (left), the original dataset exhibits several critical shortcomings: (1) insufficient video sequence length, (2) implicit inclusion of the test view within the training data. Besides, (3) the original evaluation protocol evaluates the full image rather than the dynamic regions. The above demonstrated limitations significantly bias when benchmarking dynamic reconstruction methods, ultimately compromising the validity of comparative evaluations.

Consequently, we propose the NVIDIA-LS (Yoon et al., 2020) dataset based on the original NVIDIA (Yoon et al., 2020) video sequences, as shown in Fig. 7 (right). Our pipeline begins by processing the original video sequences to obtain initial coarse camera poses from the first successful SfM (Schonberger & Frahm, 2016) timestamp, and applies consistent undistortion for all

other frames. To address the limitations of the original dataset, we implement three key improvements: (1) extending video sequences to a maximum of 160 frames (with uniform sampling applied to longer sequences), thereby increasing optimization complexity; (2) setting the first camera as the test view and all subsequent views as training data to prevent the implicit test-view inclusion during training; and (3) using SAM2 (Ravi et al., 2024) to annotate dynamic region masks for Ground Truth images of the test view, enabling more precise evaluation of dynamic reconstruction quality.

## C  IMPLEMENTATION DETAILS

### C.1  TRAINING LOSSES

Following MoSca (Lei et al., 2025) and SoM (Wang et al., 2024b), the final training loss can be represented as

$$
\mathcal{L} = \lambda_{\text{rgb}}\mathcal{L}_{\text{rgb}} + \lambda_{\text{depth}}\mathcal{L}_{\text{depth}} + \lambda_{\text{track}}\mathcal{L}_{\text{track}} \\
+ \lambda_{\text{arap}}\mathcal{L}_{\text{arap}} + \lambda_{\text{acc}}\mathcal{L}_{\text{acc}} + \lambda_{\text{vel}}\mathcal{L}_{\text{vel}},
\tag{8}
$$

where $\mathcal{L}_{\text{rgb}}$ is the photomentric regularization as the main loss, $\mathcal{L}_{\text{depth}}$ is the depth loss for regularizing the geometric structures, $\mathcal{L}_{\text{track}}$ is the tracking loss for supervising the track map with predicted 2D tracklets, $\mathcal{L}_{\text{arap}}$ is the physically-inspired As-Rigid-As-Possible (ARAP) loss for encouraging the preservation of the local distances and coordinates, $\mathcal{L}_{\text{acc}}$ and $\mathcal{L}_{\text{vel}}$ are the acceleration and velocity losses for supervising the temporal smoothness of the reconstructed deformation field.

### C.2  TRAINING DETAILS AND HYPER-PARAMETERS

The max TPT height is set to 2 (counting from 0) for all datasets, and we only optimize the camera poses at the height of 0, *i.e.* the root, of TPT for all datasets. For the optimization of the static regions, we optimize the static regions for all layers of TPT in the NVIDIA-LS (Yoon et al., 2020) dataset, and only optimize the static regions at the root of TPT in the DyCheck (Gao et al., 2022a) dataset. The weights for regularizations in Eq. 8 are mostly inherited from MoSca (Lei et al., 2025). We deactivate the tracking loss of the child-nodes for a more stable training on the DyCheck dataset. The optimization steps for each layer are set to $4000, 2000, 2000$ and $4000, 1000, 1000$, and the start point of the root for optimizing poses is set to $1500$ and $1000$ in the NVIDIA-LS (Yoon et al., 2020) dataset and DyCheck (Gao et al., 2022a) dataset, respectively. The inherited Gaussian points number from the parent node is set to $5000, 2000$ for the layers except the root. The optimization step for the stage of bundle adjustment is set to $2000$ for all datasets. The optimization step for the stage of static warm-up is set to $6000$ for all datasets. Most hyper-parameters are inherited from MoSca, and partial hyper-parameters are chosen heuristically based on the length of temporal intervals. The setting of the randomness seed for the photometric optimization is following MoSca. We select MoSca as the codebase of our method. We implement our method using PyTorch with CUDA 11.8. All experiments are conducted using RTX 3090.

### C.3  EVALUATION DETAILS

The quantitative and qualitative results of the NVIDIA-LS (Yoon et al., 2020) dataset are obtained by our reproduction. The quantitative results of the DyCheck (Gao et al., 2022a) dataset are from the reported results of MoSca (Lei et al., 2025) and RoDynRF (Liu et al., 2023). The qualitative results of MoSca (Lei et al., 2025) and RoDynRF (Liu et al., 2023) in the DyCheck (Gao et al., 2022a) dataset are from the released rendering results of them. We report the results of an individual experiment.

## D  REPRODUCTION DETAILS

**D3DGS and 4DGS.**  We inherit the configuration on the HyperNeRF (Park et al., 2021b) dataset of D3DGS (Yang et al., 2023b) and 4DGS (Wu et al., 2023) for reproducing on the NVIDIA-LS (Yoon et al., 2020) dataset. Furthermore, we provide two settings of reproduction, namely using Colmap initial points or using random initial points. For Colmap initial points, we utilize the 3D points from Colmap results of the Colmap processing on the NVIDIA-LS (Yoon et al., 2020) dataset. For random

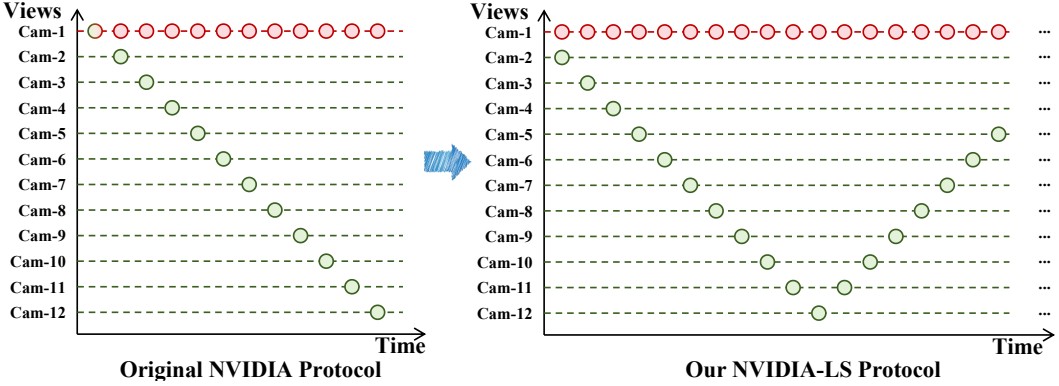

Figure 7: The protocol of the NVIDIA-LS dataset. The green points represent training data, and the red points represent testing data.

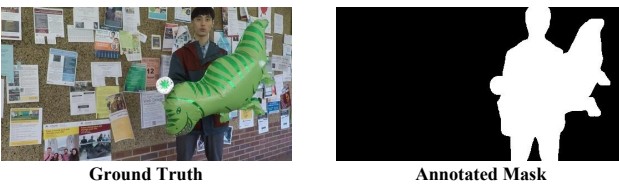

Figure 8: The example of the annotated mask for evaluation in the NVIDIA-LS dataset.

initial points, we generate random points for initialization following the strategy of DNGaussian (Li et al., 2024).

**SplineGS.** Since there are differences between datasets, some changes have to be made for reproduction. Specifically, in the original NVIDIA (Yoon et al., 2020) dataset, the test view is included in the training views as the first camera. So SplineGS (Park et al., 2025) sets the first view of optimized camera poses as the test view, served for further testing of rendering. However, in the NVIDIA-LS (Yoon et al., 2020) dataset, the test view is not included in the training views, so we supplement the test-time optimization of poses (initialized from the pose of the first training view) during testing. Besides, since the length of timestamps for the NVIDIA-LS (Yoon et al., 2020) dataset has been extended compared to the original NVIDIA (Yoon et al., 2020) dataset, the continuity of the projection mask cannot be maintained for long sequences, so we make necessary changes to improve its robustness for successful running. Meanwhile, SplineGS (Park et al., 2025) takes quite dense tracking points for tracking on the 12-frame original NVIDIA (Yoon et al., 2020) dataset. However, it is not practical for long videos. So we reduce the number of tracking points. Specifically, we start from the number of tracking points provided by Mosca (Lei et al., 2025), and employ a similar strategy of MoSca (Lei et al., 2025) to control the number of tracking points in long video sequences. Furthermore, since SplineGS (Park et al., 2025) requires manual point prompts for dynamic region pre-annotation, we provide two masks when reproducing, one is the EPI mask computed from the epipolar error maps following MoSca (Lei et al., 2025) with the threshold of 0.0001. The other is the SAM mask with our point prompts. Last, we cancel the strategy of selecting the best PSNR result during training and only use the final training result for testing.

**HiMoR.** Following HiMoR, we pre-process the NVIDIA-LS (Yoon et al., 2020) dataset through the SoM (Wang et al., 2024b) pre-processing pipeline. We make necessary changes to fit the DROID-SLAM (Teed & Deng, 2021) pre-processed data format and use default configurations for the reproduction. However, since the DROID-SLAM (Teed & Deng, 2021) does not include the pose of the test view, we supplement the test-time optimization of poses (initialized from the pose of the first training view) during testing. Besides, since the unrobustness of HiMoR on processing long-sequence uncontinuity tracks, we make several changes to improve its robustness of visibility of long-term tracking for successful running. Since HiMoR (Liang et al., 2025) requires manual point prompts for dynamic region pre-annotation, we reproduce it with two types of masks, one is the EPI mask computed from the epipolar error maps following MoSca (Lei et al., 2025) with the threshold of 0.0001 for a fair comparison. The other is the SAM mask with our annotated point

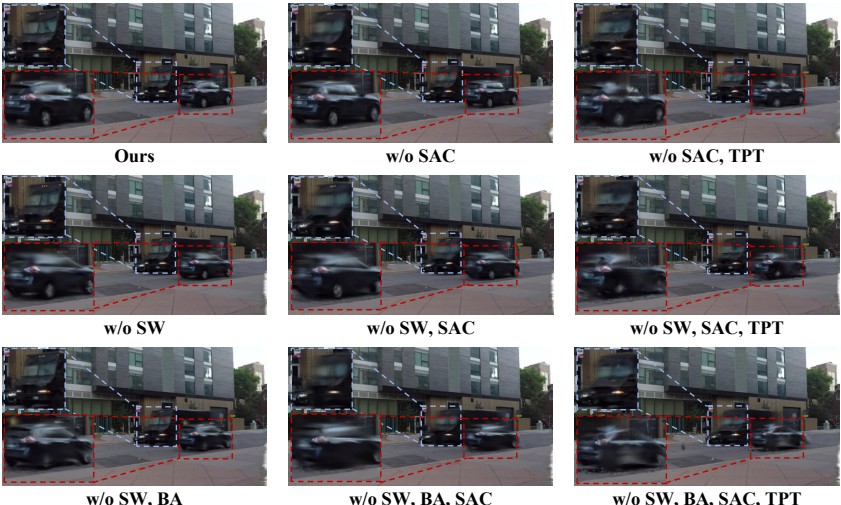

Figure 9: More qualitative ablations on the NVIDIA-LS (Yoon et al., 2020) dataset.

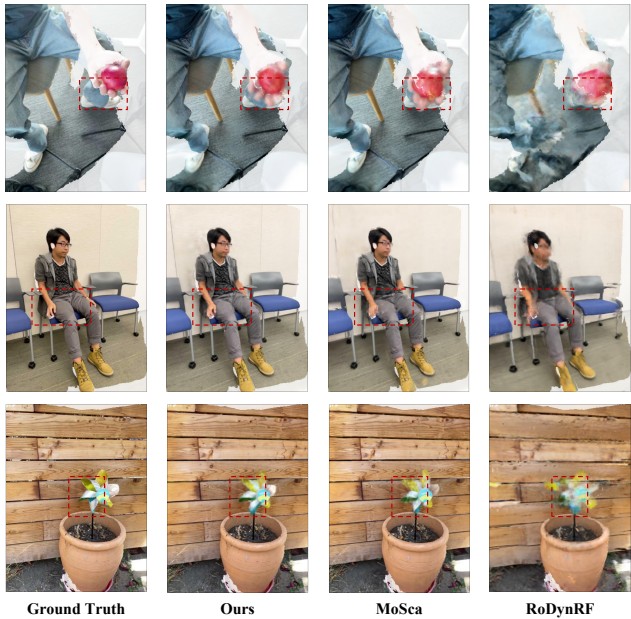

Figure 10: More qualitative comparisons on the DyCheck (Gao et al., 2022a) dataset.

prompts. It is supposed to be noted that HiMoR (Liang et al., 2025) shares the same SAM mask as SplineGS (Park et al., 2025).

**MoSca.** Since our codebase is MoSca, we only make necessary changes to MoSca to adapt to the NVIDIA-LS (Yoon et al., 2020) dataset format. The maximum number of motion nodes for node densification is set to 32 , which is identical to the experimental setting of MoSca on the Sintel (Butler et al., 2012) and Tum-Dynamics (Sturm et al., 2012) datasets.

## E    MORE RESULTS OF ABLATIONS

More detailed qualitative comparisons of ablations are shown in Fig. 9. Different from the qualitative ablation studies presented in Fig. 6 of the main manuscript, here we provide the rendering quality of different parts of the staircase elimination under various conditions in our final method. This systematic evaluation further validates the contribution of each individual component to the overall performance.

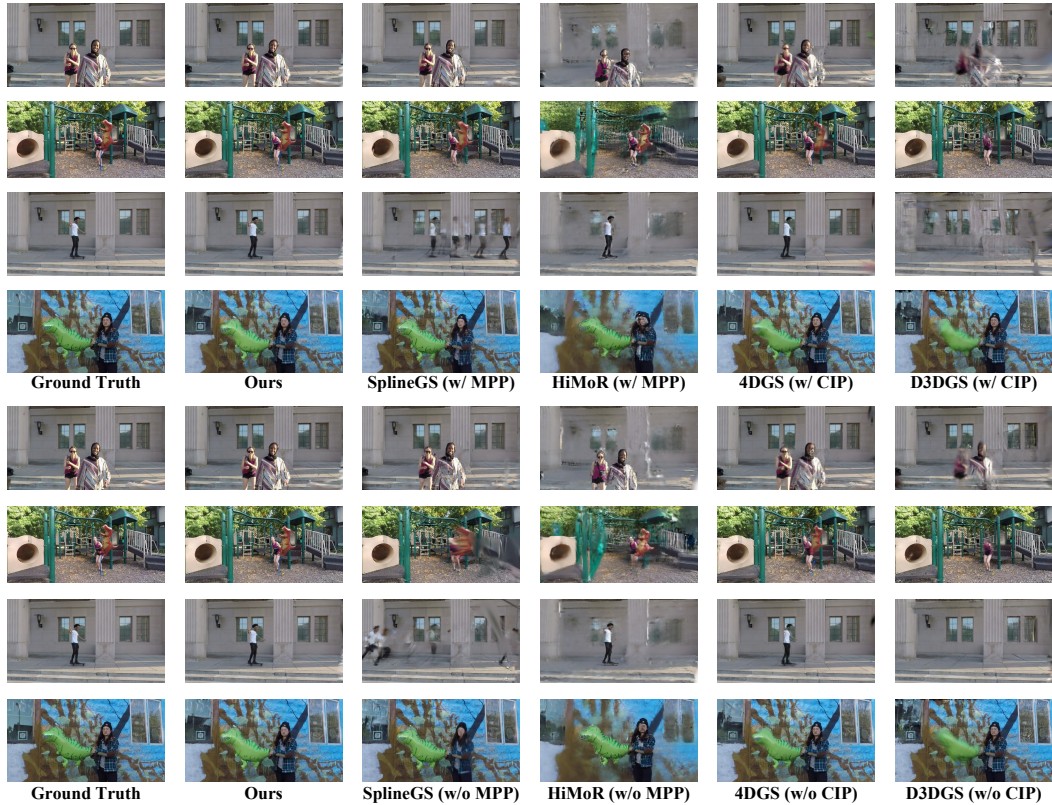

Figure 11: More qualitative comparisons on the NVIDIA-LS (Yoon et al., 2020) dataset.

## F    MORE QUALITATIVE COMPARISONS

More detailed qualitative experiments on the DyCheck (Gao et al., 2022a) dataset and NVIDIA-LS (Yoon et al., 2020) dataset are shown in Fig. 10 and Fig. 11, respectively.

## G    FAILURE CASE ANALYSIS

There remains the challenge in the reconstruction of small, fast-moving dynamic regions for our method, as well as other prevailing methods, as shown in Fig. 12. This challenge arises from weakened optimization constraints, a consequence of the inherently sparse and incomplete supervisory signals available in monocular video data. In the absence of external generative priors to supplement this limited signal, the accurate recovery of small and fast-moving motion details in these areas becomes particularly difficult. However, this is a common challenge for reconstruction-based methods under monocular constraints. In Fig. 12, we can notice that other prevailing methods exhibit more pronounced failures in such scenarios.

## H    DISCUSSION ON THE STATIC CONTENT WITHIN SCENARIOS

Our method leverages the common assumption that dynamic scenes often contain a static structural component. The SW and BA stages utilize this to robustly initialize 3D structure and camera poses. This assumption is violated in scenarios with large-scale, pervasive non-rigid deformation (*e.g.*, entire scenes deforming), which can impair initialization. However, such cases are rare in typical monocular videos, and the problem becomes severely under-constrained without static references, which is a challenge common to most methods in this situation.

Table 6: Computational cost analysis for the parallel training.

| Parallel Training (Hours) | Sequential Training (Hours) | Acceleration Ratio | Ideal Acceleration Ratio |
|---|---|---|---|
| 2.21 | 4.35 | 49.26% | 57.14% |

Table 7: Exploration of different TPT depths on the NVIDIA-LS (Yoon et al., 2020) dataset.

| Depth | mPSNR↑ | mSSIM↑ | LPIPS↓ |
|---|---|---|---|
| 0 | 17.73 | 0.655 | 0.115 |
| 1 | 18.31 | 0.681 | 0.107 |
| 2 | 18.55 | 0.692 | 0.100 |
| 3 | 18.62 | 0.695 | 0.097 |
| 4 | 18.69 | 0.698 | 0.095 |

Table 8: Exploration of different ancestral chain lengths on the NVIDIA-LS (Yoon et al., 2020) dataset.

| Length | mPSNR↑ | mSSIM↑ | LPIPS↓ |
|---|---|---|---|
| 4 | 18.69 | 0.698 | 0.095 |
| 3 | 17.44 | 0.627 | 0.117 |
| 2 | 16.55 | 0.581 | 0.128 |
| 1 | 15.54 | 0.521 | 0.143 |

## I EVALUATION OF THE PARALLEL TRAINING

We further present the corresponding computational costs analysis of the parallel training within the photometric optimization process evaluated on the scene of Balloon1, as shown in Tab. 6. In the main manuscript, we mentioned the optimization times of $O(\delta)$ for parallel training, and $O(2^{\delta+1})$ times for sequential training. Note that according to the calculation way of the Big O operator, $O(\delta) = O(\delta + 1)$, and $O(2^{\delta+1}) = O(2^{\delta+1} - 1)$. So, for the depth $\delta$ of 2, the number of nodes is 7, and the number of layers is 3, where $2 = 3 - 1$, and $7 = 2^3 - 1$. Thus, the ideal acceleration ratio is $(7 - 3)/7 \approx 57.14\%$. In practice, we achieve the acceleration ratio of $49.26\%$ because of factors such as different parameters for each layer and system I/O time, etc. However, the exploration of computational cost analysis is sufficient to demonstrate the effectiveness of parallel training.

## J FURTHER EXPLORATION ON THE TREE DEPTH AND CHAIN LENGTH

We provide further exploration on the effect of TPT depths and the ancestor chain lengths of SAC, as shown in Tab. 7 and Tab. 8. The ablation studies on the TPT depth in Tab. 7 reveal a consistent yet diminishing improvement in reconstruction quality with increased depth. This trend indicates a convergence in the representational capacity of the model. Thus, we select the depth of 2 as the trade-off between performance and computational efficiency. Furthermore, based on the reconstruction results at the depth of 4, we evaluate the results by changing the chain length during rendering, as shown in Tab. 8. The substantial performance degradation observed with reduced chains underscores the importance of hierarchical spatial information provided by SAC, validating the effectiveness of SAC in the achieved optimized reconstruction.

## K FURTHER EXPLORATION ON THE STRATEGY OF TEMPORAL SPLITTING

We additionally explore the strategy of temporal splitting for evaluating the robustness of our method, as shown in Tab. 9. Specifically, we present a systematic comparison of three different temporal splitting strategies, *i.e.*, gradient-based splitting, flow-based splitting, and our binary splitting strategy. Gradient-based splitting partitions the entire sequence by balancing the recorded gradients across intervals, whereas flow-based splitting achieves this partitioning by balancing the magnitude of optical flow within dynamic regions. We can notice that different strategies achieve similar results, demonstrating the adaptability of our method to different temporal splitting strategies. The experimental results show that our method does not depend on a specific splitting strategy but can adapt to different strategies. When adopting gradient-based or flow-based splitting strategies, their performance is only slightly different from our binary splitting strategy, confirming the robustness of our method, which stems from the inherent advantages of the hierarchical dynamic representation and optimization mechanism of our method.

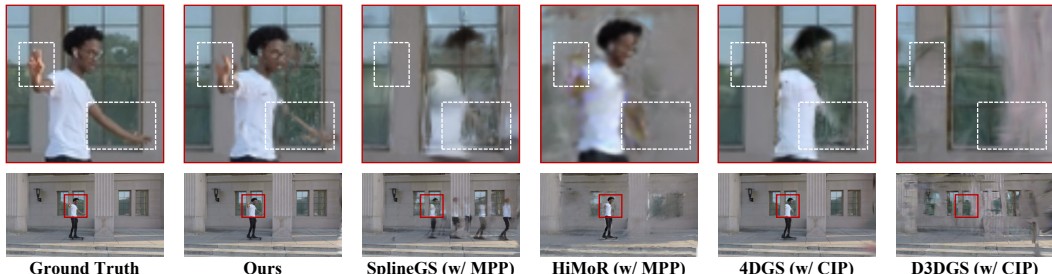



**Ground Truth**   **Ours**   **SplineGS (w/ MPP)**   **HiMoR (w/ MPP)**   **4DGS (w/ CIP)**   **D3DGS (w/ CIP)**



Figure 12: Failure case in the reconstruction of small dynamic regions with fast motion.

Table 9: Exploration on the strategy of temporal splitting.

| Settings | Metric | | | | | | |
|---|---|---|---|---|---|---|---|
| | PSNR↑ | SSIM↑ | LPIPS↓ | AVGE↓ | mPSNR↑ | mSSIM↑ | mAVGE↓ |
| Gradient-based Split | 24.04 | 0.803 | 0.100 | 0.056 | 18.54 | 0.691 | 0.092 |
| Flow-based Split | 23.98 | 0.803 | 0.101 | 0.056 | 18.46 | 0.689 | 0.093 |
| Binary Split (Ours) | 24.06 | 0.804 | 0.100 | 0.056 | 18.55 | 0.692 | 0.092 |

## L  MORE QUALITATIVE RESULTS ON THE WILD DATA

To evaluate the generalization capability of our approach to real-world, in-the-wild video sequences, we present qualitative results on the DAVIS dataset (Pont-Tuset et al., 2017). Note that we utilize DepthCrafter (Hu et al., 2024b) as depth prior and SpaTracker (Xiao et al., 2024) as tracking prior following the previous work (Lei et al., 2025). As shown in Fig. 13, we select viewpoints at timestamps corresponding to ratios of $[0, 0.25, 0.5, 0.75, 1.0]$ of the video sequence. From these selected viewpoints, we render novel views to construct a spatio-temporal matrix. This matrix provides a comprehensive visualization of the dynamics and spatial information of the scene, capturing its appearance from diverse internal viewpoints at synchronized timestamps. The results demonstrate consistent, high-fidelity renderings across all cells of this matrix. This spatial and temporal consistency indicates that our method successfully learns a robust and coherent scene representation under the wild data, which confirms the effectiveness of our method in producing accurate reconstructions from real-world video data, thereby validating its practical applicability for dynamic scene modeling.

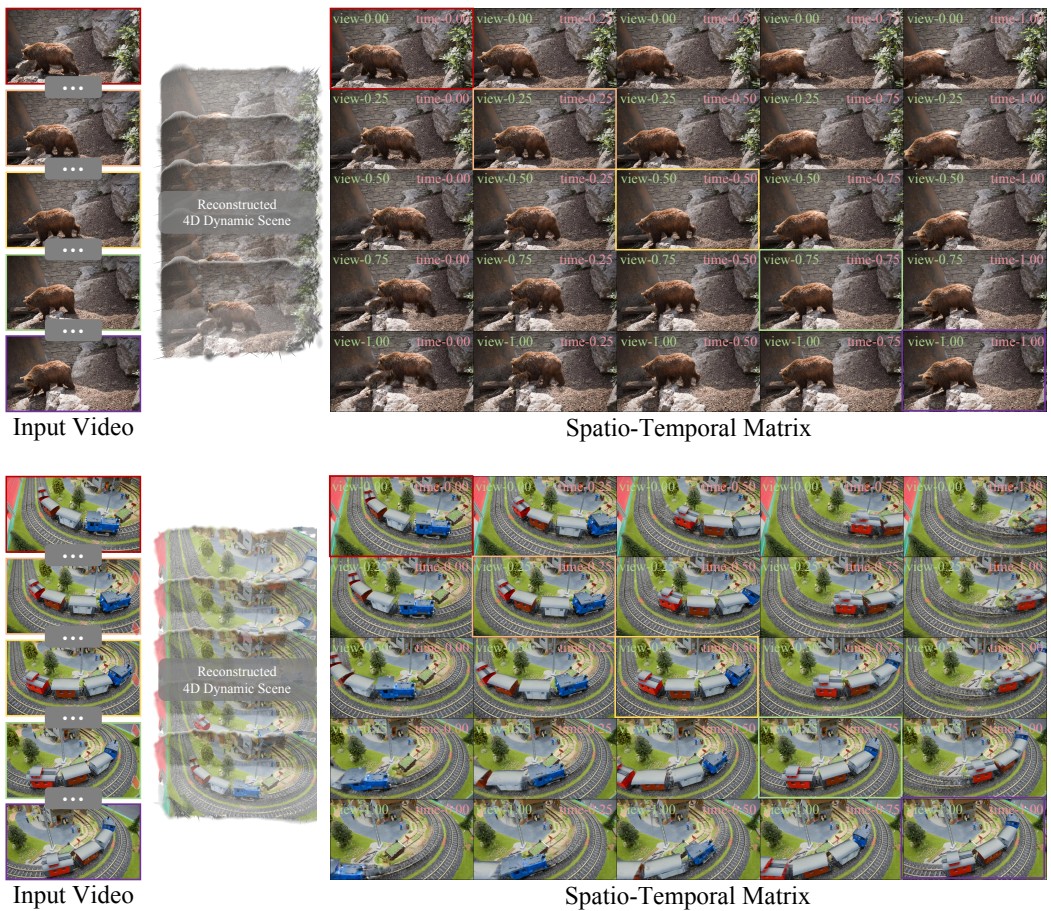

Figure 13: More qualitative results on the wild data.

