# OpenReview forum: "WorldTree: Towards 4D Dynamic Worlds from Monocular Video using Tree-Chains"
_ICLR.cc/2026/Conference — ICLR 2026 Poster_

### Official Review · Reviewer_2Tyt · 2025-10-28

**Soundness:** 2
**Presentation:** 2
**Contribution:** 2
**Rating:** 6
**Confidence:** 4

**Summary:**

The author(s) propose a unified hierarchical spatiotemporal decomposition framework for the scene to recover a 4D dynamic world from monocular video. It consists of two key components. First, a Temporal Partition Tree (TPT), which recursively splits the video’s timeline into a tree of sub-intervals (coarse-to-fine) for sequential optimization at multiple time scales, and second, a Spatial Ancestral Chains (SAC), which links each node of the time-tree with a chain of inherited scene representations from its ancestors. In essence, each time segment (node) has its own local dynamic model, while also leveraging ancestral information (broader motion context and coarse geometry from higher levels) as a complementary spatial dynamic prior. This inheritance-based design allows model specialization at different temporal scales without losing global context. The overall optimization iteratively refines the scene: starting from a root (entire video) model, then subdividing in time and re-optimizing each segment with inherited parameters in a tree structure. Notably, the tree nodes are optimized in parallel at each level, leveraging the fact that disjoint time segments can be processed independently. The SAC mechanism ensures that finer segments still incorporate multi-level spatial information by querying the chain of ancestor nodes for each segment. Qualitative results show improved and temporally coherent reconstructions of moving subjects than baselines.

**Strengths:**

1. The paper presents a hierarchical decomposition of a dynamic reconstruction problem in both time and space. Prior monocular methods generally either optimized a single global motion field over the entire video or imposed a fixed spatial hierarchy. In contrast, the paper proposes an inheritance-based time segmentation that recognizes that different temporal intervals can exhibit distinct motion patterns. Next, Spatial Ancestral Chains ensure that each temporal segment's model includes multi-scale spatial context rather than starting from scratch. This hierarchical motion specialization may help decouple global and local motions.

2.Using BFS expansion, all segments at a given depth can be optimized in parallel. Next, the use of 3D Gaussian splatting as the underlying scene representation looks like a wise choice, i.e., it provides explicit control over primitives and fast rendering.

3. The use of the SAC mechanism, i.e., aggregating the Gaussians of ancestor nodes when rendering a child's frame, effectively layers coarse and fine geometry, preventing the child from needing to re-learn large static structures or global motion from scratch.

**Weaknesses:**

1.A notable limitation is the dependence on external 2D vision models for initialization. Specifically, monocular depth estimation, optical flow, and feature tracking are used to lift 2D priors into 3D scene representations.
The paper does not show how errors in these inputs can affect the final result. Presumably, inaccuracies in initial depth or camera pose estimation could propagate through the hierarchical optimization.
Suggestion to improve: It would strengthen the work to either (a) demonstrate some robustness analysis (e.g., running the method with intentionally perturbed or lower-quality priors to see if it can recover) or (b) discuss approaches to reduce reliance on these models.

2. The TPT currently uses a very simple scheme for splitting the video (fixed). Each interval is bisected at the midpoint in time (a binary partition), and this continues to a fixed tree depth. This coarse-grained, uniform partition may not be optimal for every video. Real dynamic scenes can have uneven motion. E.g., a subject may remain still for a while then move rapidly, or different time segments have different complexity.

The paper could be improved by addressing this design choice. Perhaps providing reasoning for why uniform binary splits were chosen may be simplicity? easier parallelism?, etc.

3. WorldTree's pipeline seems to implicitly assume that the input video has a mixture of static and dynamic content, and that at least a portion of the scene can be treated as static for initialization. The authors perform a "Static Warm-Up" and Bundle Adjustment (BA) at the root stage, probably to establish camera poses and a base model of the scene before modeling non-rigid motion. This likely works well if, for example, the background is static and only an object or person is moving. The authors should clarify this assumption. Are there failure cases when the scene violates these assumptions?

**Questions:**

Q1. In Sec. 3.3, the paper states that the common ancestors of different nodes are independent of each other but have the same optimized initialization. Kindly clarify what this means in practice? For example, when a parent node (covering a certain interval) is split into two children, do you duplicate the parent's Gaussian primitives and motion parameters into each child's ancestral chain as an initialization, and then allow each branch to optimize those copies independently?

Q2. The current implementation always splits intervals evenly at the midpoint in time. Did the author(s) experiment with other criteria for the partition point? For instance, could one split an interval at a frame where the motion or reconstruction error is highest (indicating a change in dynamics)? If not, how sensitive is the method to the exact placement of the split?

---

> ### Author Response · Authors · 2025-11-23
> **Response to Reviewer 2Tyt (Part1: W1)**
>
> Thanks for the effort and patience in reviewing this work. We appreciate the valuable comments and concerns for this work. The additional experimental results and discussions have been supplemented in the revised manuscript with red marking. We will further polish the manuscript in a future version. According to the concerns, the corresponding responses are detailed below.
>
> **W1**: **The dependence on external 2D vision models for initialization. More robustness analysis and more discussions on it.**
>
> **A**: Thanks for the valuable concern regarding external 2D vision models for initialization and the potential impact of their errors. To systematically analyze the sensitivity of our method to different external priors, we have conducted additional experiments evaluating our method under different initial prior conditions, as shown in **Table 1**.
>
> The results demonstrate that while the reconstruction quality is influenced by the depth and tracking prior initialization (as is the case for other compared methods, including MoSca), the relative improvement achieved by our method remains significant and consistent. For example, when using the potentially noisier Metric3D-V2 + BootsTAPIR (lower-quality)​ configuration, our approach still achieves a significant improvement of **10.27\%** in mAVGE over Base. Similarly, under other prior combinations, such as UniDepth + CoTracker3, our method maintains a substantial gain of **12.00\%** in mAVGE. These experimental results indicate that our method does not critically depend on highly accurate initialization. The hierarchical optimization mechanism of the TPT and the complementary spatial representations from the SAC work together to progressively refine and correct inaccuracies introduced in the initial lifting step. Our method achieves robustness and effectiveness when using different external 2D vision models.
>
> Thanks again for the careful review. We have added these experimental results and the corresponding discussion to **the revised manuscript (Section 4.3 with Table 4 in the main text, Pages 9,10)**.
>
> **Table 1. Exploration of external priors on the NVIDIA-LS dataset.**
>
> | Depth Prior | Tracking Prior | Methods | mPSNR$\uparrow$ | mSSIM$\uparrow$ | mAVGE$\downarrow$ |
> |-------------|----------------|---------|---------|---------|---------|
> | UniDepth | CoTracker3 | MoSca | 18.30 | 0.674 | 0.097 |
> | UniDepth | CoTracker3 | Base | 18.17 | 0.673 | 0.100 |
> | UniDepth | CoTracker3 | Ours | **18.98** | **0.714** | **0.088** |
> | Metric3D-V2 | BootsTAPIR | MoSca | 16.94 | 0.609 | 0.142 |
> | Metric3D-V2 | BootsTAPIR | Base | 16.80 | 0.600 | 0.146 |
> | Metric3D-V2 | BootsTAPIR | Ours | **17.43** | **0.640** | **0.131** |
> | Metric3D-V2 | CoTracker3 | MoSca | 17.89 | 0.664 | 0.101 |
> | Metric3D-V2 | CoTracker3 | Base | 17.73 | 0.655 | 0.104 |
> | Metric3D-V2 | CoTracker3 | Ours | **18.55** | **0.692** | **0.092** |

---

> ### Author Response · Authors · 2025-11-23
> **Response to Reviewer 2Tyt (Part2: W2, W3)**
>
> **W2**: **The binary split may not be optimal. Perhaps providing reasoning for why uniform binary splits were chosen may be simplicity? easier parallelism?, etc.**
>
> **Related Q2**: **Experiment with other criteria for the partition point.**
>
> **A**: We thank the reviewer for the valuation concern regarding our choice of the binary splitting strategy and the potential impact of partition point selection. According to the concern, we have supplemented additional experiments to evaluate the sensitivity of our method to different splitting strategies. As shown in **Table 2**, we compare our binary splitting strategy with two adaptive alternatives: Gradient-based splitting, which selects partition points to balance gradients across child intervals, and Flow-based splitting, which selects partition points to balance optical flow magnitude of dynamic regions across child intervals.
>
> The experimental results indicate that all three strategies yield similar reconstruction quality across evaluation metrics. This consistency demonstrates that the reconstruction quality of our method is robust to the strategy of temporal split. The hierarchical coarse-to-fine optimization mechanism of the TPT, combined with the contextual compensation within hierarchical dynamic representation from the SAC, enables the model to adapt to different split points effectively. While adaptive splitting may offer benefits in some conditions, the robustness shown here suggests that the performance of our method is driven primarily by TPT and SAC, rather than the specific partitioning heuristic. Although the current results validate that our method remains robust and effective with a simple binary split, we agree that the heuristic partition strategies remain potential, and we leave it to future work.
>
> Thanks again for the effort of this review. The corresponding experimental results and discussions have been added to **the revised manuscript (Section K with Table 9 in the Appendix, Pages 20,21)**.
>
> **Table 2. Exploration on the strategy of temporal splitting.**
>
> | Settings | PSNR$\uparrow$ | SSIM$\uparrow$ | LPIPS$\downarrow$ | AVGE$\downarrow$ | mPSNR$\uparrow$ | mSSIM$\uparrow$ | mAVGE$\downarrow$ |
> |----------|--------|--------|---------|--------|---------|---------|---------|
> | Gradient-based Split | 24.04 | 0.803 | 0.100 | 0.056 | 18.54 | 0.691 | 0.092 |
> | Flow-based Split | 23.98 | 0.803 | 0.101 | 0.056 | 18.46 | 0.689 | 0.093 |
> | Binary Split (Ours) | 24.06 | 0.804 | 0.100 | 0.056 | 18.55 | 0.692 | 0.092 |
>
> **W3**: **Implicit assumption that the input video has a mixture of static and dynamic content, and that at least a portion of the scene can be treated as static for initialization. The authors should clarify this assumption.**
>
> **A**: Thanks for the efforts and patience in this reviewing. Our method does indeed leverage the common scenario where a dynamic scene contains static component. The "Static Warm-Up" and Bundle Adjustment (BA) steps are designed to exploit this prior to robustly initialize a coherent 3D structure and camera poses. But, please let us clarify that the camera trajectory is dynamic (meaning the static area is also moving in the captured images), and the static scene assumption pertains to the background geometry, not the camera viewpoint.
>
> The counterexample of the implicit assumption would arise in scenarios that severely violate the static background assumption, such as a scene undergoing large-scale, non-rigid deformation across all surfaces simultaneously,  like everything is deforming. This would consequently impair the subsequent dynamic modeling stages that build upon the initialization. This limitation is shared by most methods, as the problem becomes severely under-constrained without persistent static features, but this is rarely present in most monocular videos.
>
> Thanks again for the valuable concern. We have supplemented corresponding discussions in **the revised manuscript (Section H in the Appendix, Page 19)**.

---

> ### Author Response · Authors · 2025-11-23
> **Response to Reviewer 2Tyt (Part3: Q1, Q2)**
>
> **Q1**: **In Sec. 3.3, the paper states that the common ancestors of different nodes are independent of each other but have the same optimized initialization. Kindly clarify what this means in practice? For example, when a parent node (covering a certain interval) is split into two children, do you duplicate the parent's Gaussian primitives and motion parameters into each child's ancestral chain as an initialization, and then allow each branch to optimize those copies independently?**
>
> **A**: Thanks for the careful review. The understanding is correct. As mentioned in the paper, these initialized parameters in the ancestral chain of different child nodes are not kept tied or synchronized​ during optimization. Each child node specializes​ the inherited parameters of the ancestral chain independently based on the specific spatiotemporal content of its own temporal interval. This process enables hierarchical motion decoupling, making tree-chains learn to represent distinct motion patterns within each sub-interval. Thus, the phrase "independent of each other but have the same optimized initialization" means that common ancestry ensures coherent initial representation, while independent specialization allows for flexible and decoupled motion modeling across the hierarchy.
>
> **Q2**: **Experiment with other criteria for the partition point.**
>
> **A**: Related to **W2**. Please see the corresponding response of **W2**  (and detailed in **Table 2**).
>
> We hope our response and the additional experimental results have clarified the raised points. We are open to further questions, suggestions, or discussion to provide any needed clarification. Thanks again for the effort and patience on this review.

---

### Official Review · Reviewer_ReiE · 2025-10-31

**Soundness:** 3
**Presentation:** 3
**Contribution:** 3
**Rating:** 6
**Confidence:** 3

**Summary:**

The authors present WorldTree, a framework for 4D dynamic scene reconstruction from monocular video. It introduces a hierarchical spatiotemporal decomposition via two key components: (1) Temporal Partition Tree (TPT) – a coarse-to-fine hierarchical temporal optimization structure. (2) Spatial Ancestral Chains (SAC) – hierarchical spatial composition from ancestor nodes to decouple motion representations. These innovations allow improved performance on monocular dynamic reconstruction benchmarks (e.g., NVIDIA-LS and DyCheck) without requiring multi-view input or manual point prompts. The method shows impressive gains in mLPIPS and mPSNR and is backed by a complete ablation study.

**Strengths:**

- The paper is well-written and the results are comparable to the state-of-the-art methods if not better.
- The approach reduces the reliance on expensive external priors such as COLMAP points or manual masks, pushing towards a more practical problem setup.
- The method achieves state-of-the-art performance on the NVIDIA-LS and DyCheck benchmarks, with comparable results to methods using stronger priors.

**Weaknesses:**

- While the TPT design that uses coarse-to-fine temporal partitioning is scalable, the binary split heuristic may limit the adaptiveness in scenes with irregular motion patterns.
- It seems that the transition between subtree boundaries is not explicitly handled, which might lead to edge artifacts in the final reconstruction. It would be nice to see more details on how the method handles the transition.
- Generalization to real-world videos such as those grabbed from the Internet (or simply DAVIS dataset) is not tested.
- It is not analyzed how the external priors such as RAFT or Metric3D-v2 would affect the method performance. This might cause a problem for videos with ambiguous geometry or fast motions.

**Questions:**

- How does the method handle occlusion? Is there a single accumulated canonical representation built for the entire scene?
- Your Temporal Partition Tree (TPT) uses a fixed binary split strategy. Did you consider or experiment with adaptive or learned temporal partitioning based on motion complexity or energy? If so, how did they compare? How sensitive is the reconstruction quality to the tree depth?
- SAC inherits motion features from ancestor nodes. How does the method ensure that outdated or inaccurate ancestral representations do not propagate errors into child nodes?
- Are there common failure cases you observed (e.g., fast motion, occlusion, camera jitter)? It would be helpful to know where the method performs poorly.

---

> ### Author Response · Authors · 2025-11-23
> **Response to Reviewer ReiE (Part1: W1, W2)**
>
> Thanks for the valuable comments. We appreciate the efforts and patience in reviewing this work. The additional experimental results and discussions have been supplemented in the revised manuscript with red marking. We will further polish the manuscript in a future version. According to the concerns, the corresponded responses are detailed below.
>
> **W1**: **The binary split heuristic may limit the adaptiveness in scenes with irregular motion patterns.**
>
> **Related questions in Q2**: **Your Temporal Partition Tree (TPT) uses a fixed binary split strategy. Did you consider or experiment with adaptive or learned temporal partitioning based on motion complexity or energy?**
>
> **A**: Thanks for raising this concern regarding the adaptability of our binary splitting strategy. We agree that adaptive temporal partitioning based on motion complexity is an interesting direction. In our current design, the binary splitting heuristic was chosen for its simplicity and efficiency, but we have explicitly evaluated its robustness against other motion-aware strategies.
>
> As shown in **Table 1**, we compared our binary splitting approach with two adaptive strategies: **gradient-based splitting**, which balances gradients across child intervals, and **flow-based splitting**, which balances motion intensity across child intervals based on optical flow magnitude of dynamic regions. The results demonstrate that all three strategies achieve similar performance across all metrics. This indicates that the reconstruction quality of WorldTree is robust to the specific choice of temporal segmentation method. The hierarchical architecture of TPT, combined with the hierarchical spatial modeling of SAC, allows our framework to adapt effectively to different motion patterns. The consistent performance across strategies suggests that the strength of our approach lies not in the precise selection of split points, but in the coarse-to-fine optimization mechanism and hierarchical dynamic representation​. While adaptive or learned partitioning could be explored in future work, the current results validate that our method remains robust and effective with a simple binary split heuristic.
>
> Thanks again for the careful review. The corresponding experimental results and discussions have been supplemented in **the revised manuscript (Section K with Table 9 in the Appendix, Pages 20,21)**.
>
> **Table 1. Exploration on the strategy of temporal splitting.**
>
> | Settings | PSNR$\uparrow$ | SSIM$\uparrow$ | LPIPS$\downarrow$ | AVGE$\downarrow$ | mPSNR$\uparrow$ | mSSIM$\uparrow$ | mAVGE$\downarrow$ |
> |----------|--------|--------|---------|--------|---------|---------|---------|
> | Gradient-based Split | 24.04 | 0.803 | 0.100 | 0.056 | 18.54 | 0.691 | 0.092 |
> | Flow-based Split | 23.98 | 0.803 | 0.101 | 0.056 | 18.46 | 0.689 | 0.093 |
> | Binary Split (Ours) | 24.06 | 0.804 | 0.100 | 0.056 | 18.55 | 0.692 | 0.092 |
>
> **W2**: **The transition between subtree boundaries is not explicitly handled.**
>
> **A**: Thanks for the effort in this review. While the TPT segments the temporal sequence into subtrees for coarse-to-fine optimization, the SAC of our method ensures smooth transitions across these boundaries.  Specifically, the SAC hierarchically incorporates spatial information from ancestor nodes along the tree chain, thus achieving an implicit blending mechanism. Consequently, the representation for each leaf node is not isolated, and it is enriched with multi-scale spatial context from its parent and higher-level ancestors, whose temporal windows encompass the boundaries of their child nodes. This flow of contextual information across subtree boundaries effectively compensates for potential discontinuities, resulting in temporally coherent reconstructions without explicit edge handling. Therefore, the hierarchical dynamic representation of the SAC itself naturally mitigates the risk of artifacts at temporal partitions. Thanks again for the careful review.

---

> ### Author Response · Authors · 2025-11-23
> **Response to Reviewer ReiE (Part2: W3, W4)**
>
> **W3**: **Generalization to real-world videos is not tested.**
>
> **A**: To evaluate the generalization capability of our method on real-world videos, we have conducted additional experiments on the DAVIS dataset, and **the corresponding qualitative results are presented in the revised manuscript (Section 4.3 with Figure 6 in the main text, Page 10; Section L with Figure 13 in the Appendix, Pages 21,22)**. Specifically, we sample viewpoints at timestamps with ratios of [0, 0.25, 0.5, 0.75, 1.0] from the monocular video sequences. The novel views from these selected viewpoints are rendered, thus generating a spatio-temporal matrix that visualizes the scene dynamics from multiple viewpoints at synchronized timestamps. The high-quality and consistent renderings across this matrix demonstrate that our method successfully generalizes to in-the-wild videos with producing high-fidelity reconstruction results. These results robustly validate the practical applicability of our method. Thanks again for the valuable suggestion.
>
> **W4**: **Analysis on the performance effect of the external priors.**
>
> **A**: According to the concern, we have supplemented additional experiments of different external priors, as shown in **Table 2**​ below. The experimental results show the robustness of WorldTree when initialized with different combinations of depth and tracking priors, including settings with potential limitations such as BootsTAPIR (under fast motions or ambiguous geometry). However, the results demonstrate that while the absolute performance is influenced by the quality of the external priors (where MoSca is also being influenced), the relative improvement brought by WorldTree remains significant and consistent across all tested prior combinations. For instance, under the Metric3D-V2 + BootsTAPIR​ setting (which may be suboptimal in fast-motion scenes), our method still improves mAVGE by **10.27\%** compared to Base; while using UniDepth + CoTracker3, our method also achieves a **12.00\%** improvement in mAVGE.
>
> These experiments confirm that WorldTree does not overly rely on highly accurate initial priors. Instead, the hierarchical optimization mechanism of the TPT and the complementary spatial representations from the SAC work together to progressively correct inaccuracies introduced by imperfect priors. Therefore, even when the external prior is not perfect or is affected by fast motion or geometric ambiguity (like Metric3D-V2 + BootsTAPIR), WorldTree maintains robust performance, underscoring its adaptability and robustness to the initialization quality of external priors.
>
> Thanks again for the valuable concern. The corresponding experimental results and discussions have been supplemented to **the revised manuscript (Section 4.3 with Table 4 in the main text, Pages 9,10)**.
>
> **Table 2. Exploration of external priors on the NVIDIA-LS dataset.**
>
> | Depth Prior | Tracking Prior | Methods | mPSNR$\uparrow$ | mSSIM$\uparrow$ | mAVGE$\downarrow$ |
> |-------------|----------------|---------|---------|---------|---------|
> | UniDepth | CoTracker3 | MoSca | 18.30 | 0.674 | 0.097 |
> | UniDepth | CoTracker3 | Base | 18.17 | 0.673 | 0.100 |
> | UniDepth | CoTracker3 | Ours | **18.98** | **0.714** | **0.088** |
> | Metric3D-V2 | BootsTAPIR | MoSca | 16.94 | 0.609 | 0.142 |
> | Metric3D-V2 | BootsTAPIR | Base | 16.80 | 0.600 | 0.146 |
> | Metric3D-V2 | BootsTAPIR | Ours | **17.43** | **0.640** | **0.131** |
> | Metric3D-V2 | CoTracker3 | MoSca | 17.89 | 0.664 | 0.101 |
> | Metric3D-V2 | CoTracker3 | Base | 17.73 | 0.655 | 0.104 |
> | Metric3D-V2 | CoTracker3 | Ours | **18.55** | **0.692** | **0.092** |

---

> ### Author Response · Authors · 2025-11-23
> **Response to Reviewer ReiE (Part3: Q1, Q2, Q3, Q4)**
>
> **Q1**: **How does the method handle occlusion? Is there a single accumulated canonical representation built for the entire scene?**
>
> **A**: According to the concerns of occlusion handling and canonical representation, the corresponding illustration is detailed below.
>
> **Occlusion handling of the dynamic foregrounds**: To make the model focus more on dynamic representation during the subsequent optimization, our method first processes the static regions warm-up using the EPI masks obtained from pre-computed optical flows. This provides the stable static scene structure, which serves as a foundation for handling the dynamic occlusions.
>
> **Canonical representation**: The canonical representation of the entire scene is not "single accumulated". Instead, it is distributed across the hierarchical chains of SAC, which maintains a set of dynamic representations in the TPT. The complete canonical scene is therefore expressed as a collection of these tree-chains. The SAC chains allow the model to access multi-scale spatial information from ancestor nodes, enabling a coherent and continuous scene representation without relying on a single accumulated canonical representation.
>
> **Q2**: **Your Temporal Partition Tree (TPT) uses a fixed binary split strategy. Did you consider or experiment with adaptive or learned temporal partitioning based on motion complexity or energy? If so, how did they compare? How sensitive is the reconstruction quality to the tree depth?**
>
> **A**: For the split strategy, it is related to **W1**. Please see the corresponding response of **W1** (and detailed in **Table 1**).
>
> For the relationship between reconstruction quality and tree depth, we have conducted additional experiments to investigate the influence of the hierarchical depth of the TPT, as shown in **Table 3**.
>
> Ablation studies on TPT depth in **Table 3** demonstrate that increasing the depth leads to consistent gains in reconstruction quality. However, the magnitude of improvement gradually tapers off with each additional layer, as the model approaches its optimal reconstruction quality. Based on these observations, we selected a depth of 2 as it offers a balance between reconstruction performance and computational cost.
>
> **Table 3. Exploration on the different tree depths.**
> | Depth | mPSNR$\uparrow$ | mSSIM$\uparrow$ | LPIPS$\downarrow$ |
> |---------|---------|---------|---------|
> | 0 | 17.73 | 0.655 | 0.115 |
> | 1 | 18.31 | 0.681 | 0.107 |
> | 2 | 18.55 | 0.692 | 0.100 |
> | 3 | 18.62 | 0.695 | 0.097 |
> | 4 | 18.69 | 0.698 | 0.095 |
>
> We have supplemented the additional experimental results on the exploration of the depth of TPT in **the revised manuscript (Section J with Table 7 in the Appendix, Page 20)**. Thanks again for the careful review of this work.
>
> **Q3**: **SAC inherits motion features from ancestor nodes. How does the method ensure that outdated or inaccurate ancestral representations do not propagate errors into child nodes?**
>
> **A**: Thanks for the valuable concern. The SAC avoids simply inheriting dynamic representations from ancestor nodes. Instead, it enables each node within the hierarchy to develop a specialized representation tailored to its specific temporal scope. In this structure, ancestor nodes capture coarse and global dynamics, which child nodes then refine to represent fine-grained motions. Specializing in ancestral chains effectively addresses the hierarchical-coupling challenge within the dynamic representation of hierarchical modeling composition, ensuring that spatial information flows cohesively and accurately. Thanks again for the effort of this review.
>
> **Q4**: **Are there common failure cases you observed (e.g., fast motion, occlusion, camera jitter)? It would be helpful to know where the method performs poorly.**
>
> **A**: The limitation we observed lies in the reconstruction of small dynamic regions with fast motion. This is due to weak optimization constraints caused by insufficient supervision signals provided by monocular video. Intuitively, without external generation priors as additional supplementation, it is difficult to recover these signals. Therefore, while our method is limited in these cases, we observed potentially more severe failures in reconstructing small dynamic regions with fast motion on other prevailing reconstruction-based methods. **The corresponding qualitative results and discussion have been added to the revised manuscript (Section G with Figure 12 in the Appendix, Pages 19,21).**
>
> We hope our response and the additional experimental results have clarified the raised points. We are open to further questions, suggestions, or discussion to provide any needed clarification. Thanks again for the effort and patience on this review.

---

### Official Review · Reviewer_Dzb1 · 2025-10-31

**Soundness:** 3
**Presentation:** 3
**Contribution:** 3
**Rating:** 6
**Confidence:** 4

**Summary:**

This paper introduces WorldTree, a comprehensive framework designed to improve the efficiency and quality of 4D dynamic scene reconstruction from monocular video. The central contribution lies in its dual decomposition strategy: the Temporal Partition Tree (TPT), which facilitates coarse-to-fine temporal optimization via inheritance, and the Spatial Ancestral Chains (SAC), which handles hierarchical spatial decomposition. This structural approach directly addresses known bottlenecks in dynamic NeRF/Gaussian Splatting literature, specifically the computational overhead of holistic temporal optimization. The methodology is clever and provides a fresh perspective on spatiotemporal modeling.

**Strengths:**

1. The division of the 4D space into TPT (Temporal) and SAC (Spatial) is the most substantial contribution. The TPT’s inheritance-based optimization scheme is a highly promising avenue for reducing the redundancy and computational load associated with optimizing motion across long video sequences, potentially leading to better temporal coherence.

2. The framework explicitly aims to overcome the coupling inherent in many hierarchical methods. If the TPT successfully decouples temporal optimization, it offers a crucial step towards scaling dynamic reconstruction to very long, complex videos.

3. Committing to monocular input significantly broadens the practical applicability of the work, moving the field closer to real-world use cases where calibrated multi-view rigs are unavailable.

**Weaknesses:**

1. Dependency on External Segmentation (SAM): The reliance on external segmentation tools like SAM (mentioned in the Appendix and implied by comparisons to HiMoR/SplineGS) is a significant point of concern. If the performance gains are largely attributed to clean, pre-processed dynamic masks, the "end-to-end" nature and robustness of WorldTree in unconstrained settings are compromised. A clearer analysis is needed to quantify the degradation when using noisy or no segmentation masks.

2. Scalability and VRAM Limits: The paper notes a constraint on the maximum number of motion nodes (32) due to VRAM capacity on the NVIDIA-LS dataset. While this is a practical limitation of the current implementation (building on MoSca), it raises critical questions about the theoretical scalability of the Tree-Chains spatial representation itself. Is this limit due to the underlying MoSca architecture, or is the complexity of the Tree-Chains structure the bottleneck? This should be analyzed and discussed in the main text.

**Questions:**

See weaknesses above.

---

> ### Author Response · Authors · 2025-11-23
> **Response to Reviewer Dzb1**
>
> Thanks for the effort in reviewing this work. We appreciate the valuable comments on this work. The additional experimental results and discussions have been supplemented in the revised manuscript with red marking. We will further polish the manuscript in a future version. According to the concerns, the corresponding responses are detailed below.
>
> **W1**: Dependency on External Segmentation (SAM).
>
> **A**: Please let us kindly remind that there may exist a possible misunderstanding. We would like to clarify the issue of "Dependency on External Segmentation (SAM)".
>
> In this work, we employed SAM2 to generate corresponding dynamic foreground masks by **annotating test images**, which is adopted to better evaluate the reconstruction quality of dynamic foregrounds for the monocular 4D reconstruction. It should be noted that **SAM was not utilized during training** for this work. As indicated in Table 1 of the manuscript, our method does not rely on manual point prompts for SAM-based annotation of dynamic foreground objects.
>
> According to the concern, we have **revised the manuscript more clear of the description related to SAM (Section 4.1 in the main text, Page 7)**. Thanks again for the concern about this work.
>
> **W2**: Scalability and VRAM Limits.
>
> **A**: Thanks for the efforts of this reviewing. We appreciate the valuable comments on this work.
>
> We would like to first address a typographical inaccuracy: our method is not affected by the "maximum number of motion nodes (32)" parameter, as we disabled node densification during project development while retaining this parameter in the configuration file. Additionally, we attempted to reproduce the out-of-memory issue of MoSca when the "maximum number of motion nodes (32)" constraint was removed. Although GPU memory usage nearly approached the full 24 GB, no OOM error occurred. This discrepancy may be attributed to changes in parameters or dataset construction during the project development process. Nevertheless, removing the "maximum number of motion nodes (32)" setting did not lead to significant differences in the reproduced results of MoSca. As shown in **Table 1**, this configuration induces quite marginal deviations across multiple evaluation metrics. We have corrected the relevant description accordingly in **the revised manuscript (Section D in the Appendix, Page 18)**.
>
> **Table 1. Variants of MoSca.**
>
> | Methods | PSNR$\uparrow$ | SSIM$\uparrow$ | LPIPS$\downarrow$ | AVGE$\downarrow$ | mPSNR$\uparrow$ | mSSIM$\uparrow$ | mAVGE$\downarrow$ |
> |----------|--------|--------|---------|--------|---------|---------|---------|
> | MoSca (result of manuscript) | 23.72 | 0.792 | 0.109 | 0.060 | 17.89 | 0.664 | 0.101 |
> | MoSca* (w/o "32") | 23.67 | 0.791 | 0.110 | 0.060 | 17.91 | 0.664 | 0.101 |
>
> Furthermore, the VRAM usage of our method and MoSca is additionally monitored (using pynvml) and compared during training, as shown in **Table 2**. We can notice that our WorldTree framework demonstrates better memory efficiency compared to MoSca over different tree depths. Specifically, while the peak VRAM usage is comparable and consistently remains within the 24 GB limit for all methods, the average VRAM consumption of our method is significantly lower, and manages resources more efficiently throughout the training process.
>
> Therefore, the VRAM limitation is a constraint inherited from the underlying MoSca architecture and not a bottleneck caused by our Tree-Chains representation. As shown in Table 2, the **peak VRAM** usage is nearly the same for all methods, indicating a common constraint. However, the **average VRAM** consumption of our method is consistently and noticeably lower than that of MoSca. Therefore, the Tree-Chains structure is not the scalability bottleneck.
>
> We have additionally supplemented the corresponding results and analysis in **the revised main manuscript (Section 4.3 with Table 5 in the main text, Page 10)**. Thanks again for the careful review. We appreciate the valuable comments and suggestions.
>
> **Table 2. Further exploration on VRAM costs.**
>
> | VRAM (GB) | MoSca | Ours (Layer 0) | Ours (Layer 1) | Ours (Layer 2) |
> |----------|--------|--------|---------|---------|
> | Peak VRAM | 23.99 | 23.99 | 23.99 | 24.00 |
> | Average VRAM | 12.90 | 8.61 | 11.99 | 12.68 |
>
> We hope our response and the additional experimental results have clarified the raised points. We are open to further questions, suggestions, or discussion to provide any needed clarification. Thanks again for the efforts and patience on this review.

---

### Official Review · Reviewer_ecpZ · 2025-10-31

**Soundness:** 3
**Presentation:** 3
**Contribution:** 2
**Rating:** 4
**Confidence:** 4

**Summary:**

This paper proposes a approach for monocular dynamic reconstruction. The core contributions consists of two components:
(1). Temporal Partition Tree (TPT): a hierarchical binary partitioning of the video time sequence. It can be used to refine the deformation modeling in temporal segments.
(2). Spatial Ancestral Chains (SAC): a mechanism that propagates Gaussian representations from ancestor nodes to descendant nodes through ancestral chains, which can alleviate information loss during Gaussian point inheritance.

Additionally, this paper contributes an enhanced dataset based on NVIDIA-LS.

**Strengths:**

For the experimental results, the comparisons as well as the ablation studies support the paper’s claims. For the presentation, the writing is generally clear, for example, the paper provides a comparison diagram with previous works, and the ablation study results are presented comprehensively.

**Weaknesses:**

For the experiments, the paper mentions an improvement in computational efficiency, but it lacks corresponding experimental data on computational time cost.
From the perspective of novelty, the main contributions of this paper are TPT and SAC. TPT is used for video segmentation, while SAC supplements the current information by reusing information from higher-level ancestors. However, this does not constitute a significant theoretical breakthrough. On the other hand, the proposed mechanisms rely on strong assumptions, such as reasonable temporal segmentation and proper root node initialization. Therefore, these characteristics could also become new limitations of this paper (compared with methods that directly impose various monocular video priors onto 3D Gaussians[1]).

[1] D. Wu, F. Liu, Y.-H. Hung, Y. Qian, X. Zhan, and Y. Duan, “4d-fly: Fast 4d reconstruction from a single monocular video,” in Proceedings of the IEEE/CVF Conference on Computer Vision and Pattern Recognition (CVPR), June 2025, pp. 16 663–16 673.

**Questions:**

(1). Could you provide more experimental validation on how the hierarchical depth of TPT and the ancestor chain length of SAC affect performance?
(2). If the video is very long, would the binary partitioning in TPT result in an overly deep tree? What impact would this have on performance?
(3). Does the ancestor chain in SAC introduce redundant information? Has this potential issue been considered?

---

> ### Author Response · Authors · 2025-11-23
> **Response to Reviewer ecpZ (Part1: W1, W2)**
>
> Thanks for the careful review of this work. We appreciate the efforts and valuable comments of this reviewing. The additional experimental results and discussions have been supplemented in the revised manuscript with red marking. We will further polish the manuscript in a future version. According to the concerns, the corresponded responses are detailed below.
>
> **W1**: **Computational costs analysis of the parallel training.**
>
> **A**: According to the concern, we provide corresponding computational costs analysis of the parallel training within the photometric optimization process, as shown in **Table 1**. The illustration is detailed below.
>
> In the main manuscript, we mentioned the optimization times of $O(\delta)$ for parallel training, and $O(2^{\delta + 1})$ times for sequential training. Note that according to the calculation way of the Big O operator, $O(\delta) = O(\delta + 1)$, and $O(2^{\delta + 1}) = O(2^{\delta + 1} - 1)$. So, for the depth $\delta$ of $2$, the number of nodes is $7$, and the number of layers is $3$, where $2 = 3 - 1$, and $7 = 2^3 - 1$. So, the ideal acceleration ratio is $(7-3)/7 \approx 57.14\\%$. In practice, we evaluated the computational time on "Balloon1" of NVIDIA-LS and achieved the acceleration ratio of **49.26\%**, because of factors such as different parameters for each layer and system I/O time, etc. However, the exploration of computational cost analysis is sufficient to demonstrate the effectiveness of parallel training.
>
> We also supplemented the corresponding results in **the revised manuscript  (Section I with Table 6 in the Appendix, Page 20)**. Thanks again for the careful review and suggestions.
>
> **Table 1. Computational cost of the parallel training.**
> | Parallel Training (Hours) | Sequential Training (Hours) | Acceleration Ratio | Ideal Acceleration Ratio |
> |-|-|-|-|
> | $2.21$ | $4.35$ | $49.26\\%$ | $57.14\\%$ |
>
> **W2**: **Novelty of this work.**
>
> **A**:​ Thanks for the effort in this review. According to the concern, please let us reclarify the novelty of this work, which provides a fundamental framework for monocular dynamic reconstruction.
>
> * **Challenges**: **A review of Section 1**. The prevailing methods are limited by the fundamental spatiotemporal coupling dilemma: they either enforce an entire temporal optimization that cannot adapt to varying deformation patterns, or they adopt global spatial fusion but resulting in an amorphous dynamic representation, or they introduce hierarchical spatial representations that suffer from optimization conflicts. WorldTree breaks this dilemma by proposing a unified tree-chains framework.
> * **WorldTree**: **It's not Tree and Chains. It's Tree-Chains.** WorldTree introduces a cohesive structure that facilitates temporal-wise hierarchical optimization alongside spatial-wise hierarchical representation. The TPT serves as the temporal scaffolding, constructing a multi-scale hierarchy through inheritance-based partitioning to enable specialized, coarse-to-fine optimization within intervals. This tree structure, in turn, enables the SAC to recursively infuse each node with ancestral dynamics. SAC specializes and queries these dynamics along the ancestral chains, thereby providing the complementary spatial context that TPT alone lacks and effectively preventing node isolation. The synergy between TPT and SAC establishes a dynamic information hierarchy, therefore constructing a hierarchical dynamic representation​ that is capable of handling varying deformation patterns, while simultaneously decoupling the optimization conflict, which are challenges that existing methods tough to resolve.
> * **Experimental validation**: **Tree good. Tree-Chains better.** The experimental results strongly validate this theoretical advancement.​ Our method achieves state-of-the-art performance, e.g., an **8.26\%​** improvement in LPIPS on the challenging NVIDIA-LS dataset and a **9.09\%** improvement in mLPIPS on DyCheck compared to the second-best method. The ablation studies further confirm the synergistic effect, where using TPT alone improved LPIPS by **8.70\%** and adding SAC provided a further substantial boost of improvements, increasing to **13.04\%**. These gains stem directly from the effective cooperation mechanism between TPT and SAC, demonstrating the significant effectiveness of the WorldTree framework.
>
> We hope this clarification better highlights the novelty and theoretical contribution of this work.

---

> ### Author Response · Authors · 2025-11-23
> **Response to Reviewer ecpZ (Part2: W3)**
>
> **W3**: **Dependence on assumptions like reasonable temporal segmentation and proper root node initialization.**
>
> **A**: According to the concern, the clarification is detailed below.
>
> First, thanks for pointing out the related work 4D-Fly [A] for the monocular 4D dynamic reconstruction. While 4D-Fly presents an efficient framework for streamable 4D modeling from monocular video, our WorldTree framework introduces fundamental innovations in spatiotemporal modeling that address core limitations more systematically. Specifically, whereas 4D-Fly adopts an anchor-based propagation strategy for incremental dynamic representation, it lacks an explicit mechanism for coarse-to-fine temporal partition optimization across the entire sequence. In contrast, our proposed TPT  enables hierarchical, coarse-to-fine optimization of varying deformation patterns. Moreover, 4D-Fly does not explicitly model hierarchical spatial information, a capability provided by our SAC, which offers spatial guidance for complex dynamic scenes. The discussion of 4D-Fly [A] has been incorporated into **the revised manuscript (Section 2 in the main text, Page 3)**.
>
> Furthermore, please let us clarify the comment about the proposed method relying on strong assumptions. The mentioned assumptions include two parts: **reasonable temporal segmentation** and **proper root node initialization**.
>
> For the assumption of **reasonable temporal segmentation**, we additional supplement further exploration on the strategy of temporal segmentation/splitting for evaluating the robustness of this work, as shown in **Table 2**. Specifically, **Table 2** presents a systematic comparison of three different temporal segmentation strategies: gradient-based splitting, flow-based splitting, and our binary splitting strategy. Gradient-based splitting partitions the entire sequence by balancing the recorded gradients across child intervals, whereas flow-based splitting achieves this partitioning by balancing the magnitude of optical flow within dynamic regions. We can notice that **different strategies achieve similar results**, demonstrating the WorldTree framework's adaptability to different temporal segmentation strategies. This clarifies the concerns about the "reasonable temporal segmentation" assumption. The experimental results show that the WorldTree framework does not depend on a specific segmentation strategy but can adapt to different strategies. When adopting gradient-based or flow-based segmentation strategies, their performance is only slightly different from our binary splitting strategy, practically validating the method's robustness, which stems from the inherent advantages of WorldTree's hierarchical architecture. While our method employs a binary splitting strategy, its core innovation lies in its hierarchical dynamic representation and optimization mechanism rather than specific splitting point selection.
>
> The corresponding additional experimental results have been supplemented to **the revised manuscript (Section K with Table 9 in the Appendix, Pages 20,21)**.
>
> **Table 2. Exploration on the strategy of temporal splitting.**
>
> | Settings | PSNR$\uparrow$ | SSIM$\uparrow$ | LPIPS$\downarrow$ | AVGE$\downarrow$ | mPSNR$\uparrow$ | mSSIM$\uparrow$ | mAVGE$\downarrow$ |
> |----------|--------|--------|---------|--------|---------|---------|---------|
> | Gradient-based Split | 24.04 | 0.803 | 0.100 | 0.056 | 18.54 | 0.691 | 0.092 |
> | Flow-based Split | 23.98 | 0.803 | 0.101 | 0.056 | 18.46 | 0.689 | 0.093 |
> | Binary Split (Ours) | 24.06 | 0.804 | 0.100 | 0.056 | 18.55 | 0.692 | 0.092 |

---

> ### Author Response · Authors · 2025-11-23
> **Response to Reviewer ecpZ (Part2: W3, continued)**
>
> For the assumption of **proper root node initialization**, the ablation experiments in **Table 3 of the main manuscript** show that even with BA and SW disabled (**improper root node initialization**), our method still achieves significant performance improvements. Specifically, when both BA and SW are disabled, the full method still improves the LPIPS metric by **18.71\%** compared to the baseline. Furthermore, we additionally supplement further experimental results of using different external priors as shown in **Table 3** below. Based on the additional external prior exploration experimental results shown in **Table 3** below, we can clearly observe that the WorldTree method exhibits excellent effectiveness and robustness under initialization with different combinations of depth and tracking priors. For example, the mAVGE is improved by **12.00\%** under UniDepth+CoTracker3, **10.27\%** under Metric3D-V2+BootsTAPIR, and **11.54\%** under Metric3D-V2+CoTracker3 (our original setting in the manuscript), compared to Base. This demonstrates that even under suboptimal external prior initialization conditions (like BootsTAPIR results in **improper root node initialization**), WorldTree can still effectively improve reconstruction quality through its inherent TPT and SAC mechanisms.
>
> The corresponding additional experimental results have been supplemented to **the revised manuscript (Section 4.3 with Table 4 in the main text, Pages 9,10)**.
>
> Thanks again for the careful review.
>
> **Table 3. Exploration of external priors on the NVIDIA-LS dataset.**
>
> | Depth Prior | Tracking Prior | Methods | mPSNR$\uparrow$ | mSSIM$\uparrow$ | mAVGE$\downarrow$ |
> |-------------|----------------|---------|---------|---------|---------|
> | UniDepth | CoTracker3 | MoSca | 18.30 | 0.674 | 0.097 |
> | UniDepth | CoTracker3 | Base | 18.17 | 0.673 | 0.100 |
> | UniDepth | CoTracker3 | Ours | **18.98** | **0.714** | **0.088** |
> | Metric3D-V2 | BootsTAPIR | MoSca | 16.94 | 0.609 | 0.142 |
> | Metric3D-V2 | BootsTAPIR | Base | 16.80 | 0.600 | 0.146 |
> | Metric3D-V2 | BootsTAPIR | Ours | **17.43** | **0.640** | **0.131** |
> | Metric3D-V2 | CoTracker3 | MoSca | 17.89 | 0.664 | 0.101 |
> | Metric3D-V2 | CoTracker3 | Base | 17.73 | 0.655 | 0.104 |
> | Metric3D-V2 | CoTracker3 | Ours | **18.55** | **0.692** | **0.092** |
>
> [A] Diankun Wu, Fangfu Liu, Yi-Hsin Hung, Yue Qian, Xiaohang Zhan, and Yueqi Duan. 4d-fly: Fast 4d reconstruction from a single monocular video. In Proceedings of the Computer Vision and Pattern Recognition Conference, pp. 16663–16673, 2025.

---

> ### Author Response · Authors · 2025-11-23
> **Response to Reviewer ecpZ (Part3: Q1, Q2, Q3)**
>
> **Q1**: **Could you provide more experimental validation on how the hierarchical depth of TPT and the ancestor chain length of SAC affect performance?**
>
> **A**: Here we provide further exploration on the effect of hierarchical depth of TPT and the ancestor chain length of SAC, as shown in **Table 4** and **Table 5**.
>
> The ablation studies on the TPT depth in **Table 4** reveal a consistent yet diminishing improvement in reconstruction quality with increased depth. This trend indicates a convergence in the model's representational capacity, leading us to select a depth of 2 as the trade-off between performance and computational efficiency. Furthermore, based on the reconstruction results at the depth of 4, we evaluated the results by changing the chain length during rendering, as shown in **Table 5**. We can notice that the performance is sensitive to SAC chain length. The significant performance drop due to reduced chain length highlights the importance of hierarchy space context and validates the effectiveness of SAC in the achieved optimized reconstruction.
>
> We have supplemented the additional experimental validation to **the revised manuscript (Section J with Tables 7 and 8 in the Appendix, Page 20)** on the exploration of the hierarchical depth of TPT and the ancestor chain length of SAC. Thanks again for the careful review on this work.
>
> **Table 4. Exploration on the different tree depths.**
> | Depth | mPSNR$\uparrow$ | mSSIM$\uparrow$ | LPIPS$\downarrow$ |
> |---------|---------|---------|---------|
> | 0 | 17.73 | 0.655 | 0.115 |
> | 1 | 18.31 | 0.681 | 0.107 |
> | 2 | 18.55 | 0.692 | 0.100 |
> | 3 | 18.62 | 0.695 | 0.097 |
> | 4 | 18.69 | 0.698 | 0.095 |
>
> **Table 5. Exploration on the different chain lengths.**
> | Length | mPSNR$\uparrow$ | mSSIM$\uparrow$ | LPIPS$\downarrow$ |
> |---------|---------|---------|---------|
> | 4 | 18.69 | 0.698 | 0.095 |
> | 3 | 17.44 | 0.627 | 0.117 |
> | 2 | 16.55 | 0.581 | 0.128 |
> | 1 | 15.54 | 0.521 | 0.143 |
>
> **Q2**: **If the video is very long, would the binary partitioning in TPT result in an overly deep tree? What impact would this have on performance?**
>
> **A**: Thanks for the valuable concern of this work. The binary partitioning in the TPT is designed with a predefined maximum depth precisely to prevent the scenario of an overly deep tree, even with very long video sequences. By the way, we can also directly extend the tree from the optimized model without re-training. For the exploration on the extended depth of the tree, please see the response of **Q1** (and detailed in **Table 4**).
>
> **Q3**: **Does the ancestor chain in SAC introduce redundant information? Has this potential issue been considered?**
>
> **A**: Thanks for this insightful question. The ancestor chain in our SAC is designed not to introduce redundant information, but to capture complementary multi-scale contextual information. According to the concern, our method specializes the motion representation of each ancestral node to achieve hierarchical motion decoupling. The chain progressively incorporates features from ancestral nodes, each providing additional spatial context that is unavailable at the finer, leaf-level nodes. This hierarchical integration is crucial for guiding the reconstruction of complex geometries and textures, as demonstrated in **Table 5**. Ablations show that reducing this chain leads to a substantial drop in quality, confirming that the ancestral information is indeed complementary and non-redundant.
>
> We hope our response and the additional experimental results have clarified the raised points. We are open to further questions, suggestions, or discussion to provide any needed clarification. Thanks again for the effort and patience on this review.

---

### Author Response · Authors · 2025-12-03
**Summary**

We appreciate all ACs and reviewers for their effort and their valuable concerns and suggestions in this review, which help us improve the quality of this paper further.

We have **comprehensively responded to *all* the reviewers' concerns** through detailed clarifications, expanded discussions, additional experimental analyses, and corresponding revisions of the updated manuscript. The current status of this reviewing process is:
* **Reviews** from reviewers: Yes.
* **Responses** and **Revisions** from authors: Yes.
* **Discussions**: No.

Thus, we provide a brief summary of the current reviewing process to assist the final justification of this work.

First, we appreciate the positive comments from reviewers:

* **Positive Points**
  * **R1-ecpZ** acknowledged this work that
    * *"... the comparisons as well as the ablation studies support the paper’s claims"*;
    * *"... the ablation study results are presented comprehensively"*;
  * **R2-Dzb1** acknowledged this work that
    * *"... directly addresses known bottlenecks in dynamic NeRF/Gaussian Splatting literature..."*;
    * *"... is clever and provides a fresh perspective on spatiotemporal modeling"*;
  * **R3-ReiE** acknowledged this work that
    * *"... achieves state-of-the-art performance..."*;
    * *"... pushing towards a more practical problem setup"*;
  * **R4-2Tyt**: acknowledged this work that
    * *"... help decouple global and local motions"*;
    * *"... effectively layers coarse and fine geometry..."*;

Furthermore, we also appreciate the concerns from reviewers, and the corresponding revisions of the updated manuscript are:

* **Summary of Revisions**
  * **Computational cost analysis**. Further evaluation on VRAM costs and parallel training: Section 4.3 with Table 5 in the main text (Related to R2-Dzb1), Section I with Table 6 in the Appendix (Related to R1-ecpZ).
  * **Effectiveness analysis**. Further evaluation on tree depths and chain lengths: Section J with Tables 7 and 8 in the Appendix (Related to R1-ecpZ, R3-ReiE)
  * **Robustness analysis**. Further evaluation on the different external priors: Section 4.3 with Table 4 in the main text (Related to R1-ecpZ, R3-ReiE, R4-2Tyt). Further evaluation on the strategies of temporal splitting: Section K with Table 9 in the Appendix (Related to R1-ecpZ, R3-ReiE, R4-2Tyt).
  * **Qualitative analysis**. Further evaluation on the wild data: Section 4.3 with Figure 6 in the main text, Section L with Figure 13 in the Appendix (Related to R3-ReiE). Further failure case analysis: Section G with Figure 12 in the Appendix (Related to R3-ReiE)
  * **Discussions and polishings**. Supplementation of related work: Section 2 in the main text (Related to R1-ecpZ). Clearer description related to SAM: Section 4.1 in the main text (Related to R2-Dzb1). Further discussions on the static content within scenarios: Section H in the Appendix (Related to R4-2Tyt)

Thanks again for the effort and patience in this review. We appreciate the valuable comments and suggestions. The detailed, respectful responses are presented below.

---

### Meta-Review · Area_Chair_DYuY · 2026-01-06

**Summary:**

The primary concerns raised by reviewers centered on the technical novelty of the proposed Tree-Chains framework, its practical limitations, and the robustness of its empirical validation. Specifically, reviewers questioned whether the core components (TPT and SAC) constituted a significant theoretical advance or were an incremental application of existing hierarchical and autoregressive concepts. Practical concerns included the method's dependence on external priors (e.g., depth estimation, optical flow), its computational efficiency and VRAM scalability, the adaptability of its fixed binary temporal splitting strategy, and its generalization capability to in-the-wild videos. The validity of comparisons and the need for more ablation studies (e.g., on tree depth, chain length) were also noted.

**Reviewer Concerns:**

The rebuttal effectively addressed the majority of concrete, testable concerns. The authors provided extensive new experiments analyzing VRAM costs and parallel training efficiency, robustness to different external priors (showing consistent gains even with suboptimal inputs), the impact of tree depth and chain length, and generalization results on the DAVIS dataset. They also clarified misunderstandings, such as the role of SAM (used only for evaluation, not training) and corrected a configuration parameter issue. The concerns regarding the fixed binary split were mitigated with experiments showing comparable performance using alternative gradient-based or flow-based splitting strategies. The outstanding concerns are primarily subjective: the perceived incremental nature of the technical contribution and the theoretical novelty of the unified framework, which the rebuttal argues for but may not fully convince all reviewers. The assumption of a static scene component remains a methodological limitation but is acknowledged as common in the field.

**Reviewer Scores:**

**Reviewer ecpZ (initial score: 4)**: The reviewer's main concerns about ablation studies (tree depth, chain length) and robustness to priors were directly addressed with new experiments and data. However, there is no response from Reviewer ecpZ.

**Reviewer Dzb1 (initial score: 6)**: The clarifications regarding SAM dependency and the provided VRAM analysis directly alleviated the two major weaknesses they identified.

**Reviewer ReiE (initial score: 6)**: The rebuttal comprehensively addressed all four weaknesses (split strategy, transitions, real-world testing, prior analysis) and provided thorough answers to all questions, including failure cases.

**Reviewer 2Tyt (initial score: 6)**: The new experiments on prior robustness and split strategy, along with clear answers to their questions, effectively mitigated their primary concerns about external dependencies and heuristic design choices.

---

### Decision · Program_Chairs · 2026-01-26

Accept (Poster)